# Federated Learning and EEL-Levy Optimization in CPS ShieldNet Fusion: A New Paradigm for Cyber–Physical Security

**DOI:** 10.3390/s25123617

**Published:** 2025-06-09

**Authors:** Nalini Manogaran, Yamini Bhavani Shankar, Malarvizhi Nandagopal, Hui-Kai Su, Wen-Kai Kuo, Sanmugasundaram Ravichandran, Koteeswaran Seerangan

**Affiliations:** 1Department of Computer Science and Business Systems, S.A. Engineering College (Autonomous), Chennai 600077, Tamil Nadu, India; drnalini@saec.ac.in; 2Department of Networking and Communications, School of Computing, College of Engineering and Technology, SRM Institute of Science and Technology (SRMIST), Kattankulathur 603203, Tamil Nadu, India; yamini.subagani@gmail.com; 3Department of Computer Science and Engineering, Vel Tech Rangarajan Dr. Sagunthala R&D Institute of Science and Technology, Avadi, Chennai 600062, Tamil Nadu, India; drnmalarvizhi@veltech.edu.in; 4Department of Electrical Engineering, Smart Machinery and Intelligent Manufacturing Research Center, National Formosa University, Huwei, Yunlin County 632, Taiwan; 5Department of Electro-Optics Engineering, National Formosa University, Huwei, Yunlin County 632, Taiwan; wkkuo@nfu.edu.tw (W.-K.K.); rsanmu88@gmail.com (S.R.); 6Department of CSE (Artificial Intelligence and Machine Learning), S.A. Engineering College (Autonomous), Chennai 600077, Tamil Nadu, India; s.koteeswaran@ieee.org

**Keywords:** cyber–physical systems (CPSs), security, federated learning, optimization, artificial intelligence (AI), cyber threats

## Abstract

As cyber–physical systems are applied not only to crucial infrastructure but also to day-to-day technologies, from industrial control systems through to smart grids and medical devices, they have become very significant. Cyber–physical systems are a target for various security attacks, too; their growing complexity and digital networking necessitate robust cybersecurity solutions. Recent research indicates that deep learning can improve CPS security through intelligent threat detection and response. We still foresee limitations to scalability, data privacy, and handling the dynamic nature of CPS environments in existing approaches. We developed the CPS ShieldNet Fusion model as a comprehensive security framework for protecting CPS from ever-evolving cyber threats. We will present a model that integrates state-of-the-art methodologies in both federated learning and optimization paradigms through the combination of the Federated Residual Convolutional Network (FedRCNet) and the EEL-Levy Fusion Optimization (ELFO) methods. This involves the incorporation of the Federated Residual Convolutional Network into an optimization method called EEL-Levy Fusion Optimization. This preserves data privacy through decentralized model training and improves complex security threat detection. We report the results of a rigorous evaluation of CICIoT-2023, Edge-IIoTset-2023, and UNSW-NB datasets containing the CPS ShieldNet Fusion model at the forefront in terms of accuracy and effectiveness against several threats in different CPS environments. Therefore, these results underline the potential of the proposed framework to improve CPS security by providing a robust and scalable solution to current problems and future threats.

## 1. Introduction

Artificial intelligence (AI) is central to generating value in CPSs, which are systems containing computing and physical elements. Thus, our society is primarily defined by socio-economic values as they determine the activities and objectives of various organizations [1,2]. To be more precise, AI’s capability in dealing with massive amounts of data and the ability of the latter to perform highly complex computations in a short time are of particular value here. They can process data rapidly. This means results will be generated and the organization will be able to make decisions within a short time. Since value creation is steeped in cyber–physical systems, autonomous AI will form the epicenter of economic and societal decisions [3,4]. It is evident that AI’s partnership with value creation is mutually beneficial, indicating that AI is a key influencer of society’s future. Other key paradigms useful in comprehending interconnected physical objects for communication and control are based on IoT and cyber–physical systems. IoT is defined as physical objects with embedded technologies such as sensors, software, and other systems that facilitate information sharing with other devices [5]. The two technologies have enhanced IoT-CPS advancements [6,7,8] due to wireless technology advancements. Bluetooth sensors, PDA, and other smart terminals are currently applied to numerous civil and military fields. These sensors allow the device to do a specific job or multiple jobs by sensing information and sending it [9].

Moreover, CPSs are built through the amalgamation of communication [10], computing, and control engineering capabilities, which allows these systems to undertake data-driven processes. This architecture has to be one that supports high-QoS and QoE for end users to make intelligent and context-aware decisions [11]. In other words, confidentiality ensures that only authorized personnel access specific and often sensitive information, integrity equally ensures that information is not corrupted in any way, and availability provides assurance that information can be accessed, especially during an emergency or at any time it may be required [12]. This will involve the management of accessible technologies, for example, machine learning and artificial intelligence, to enhance the performance of the network plus the allocation of resources available [13,14].

Recently, the vulnerability of CPSs to information attacks has increased because of their linkage to cyberspace. This has grown in frequency and complexity with automated attacking tools developed and professional hacking groups [15,16]. There are several reasons why deep learning (DL) [17] is much more effective than conventional machine learning (ML) methods, especially when there are lots of data available. When compared to traditional standards, DL models can effectively analyze complicated patterns and provide quality results in different domains. Nevertheless, deep learning for CPS cybersecurity has not kept pace with its adoption in other sectors. Many recent studies featured many deep learning models designed for CPS cyberattack detection. However, the practical implementation has always been hindered by the inherent difficulty of integrating such security measures into CPSs. This is due to their complexity, which makes it difficult to integrate cybersecurity without affecting their functionality [18,19].

One common thought is that applying conventional cybersecurity solutions directly to complex CPSs that often combine real-time operations comprising both physical and digital aspects is a challenge. Additionally, the continuous and interrelated character of CPSs necessitates an advanced and adaptive security approach that deals with modern threats [20,21]. Deep learning brings a potential solution by providing strong methods for recognizing and reducing cyberattacks. But this requires an understanding of what makes this system special as well as its weaknesses. The majority of AI-based techniques that are used in the production of a CPS have several problems that put their effectiveness in handling cybersecurity problems at risk [22]. Furthermore, the interpretability and explainability of conventional deep learning models contribute to the fact that people are often clueless and the explanations for the decisions are challenging to pinpoint as to the mistakes they made [23,24]. Also, deep learning algorithms typically consume substantial computation, which can slow things down and break or exclude many real-time scenarios as it causes extra latency delays and affects the time to provision a model from scratch or update an existing model quickly, adapted to a changing cybersecurity environment.

CPS is the seamless integration of computational algorithms with physical processes. This constitutes the backbone of modern infrastructures, such as smart grids, autonomous transportation, industrial automation, healthcare monitoring. CPSs are pursued for enhancement due to their ability to transform operational efficacy, decision-making accuracy, and real-time responsiveness in critical spheres. On the flip side, increasing interconnectivity and dependence on digital communication have put these systems under enormous threat from various cyberattacks, which could have terrible consequences if they are not well protected. This dual nature of CPS integrates physical assets with cyber elements and provides unique challenges in security, privacy, and resilience. As such, it has become very topical for research to develop new models that are capable of monitoring CPS intelligently, detecting attacks, and mitigating them while keeping functional integrity and data privacy intact. The motivation to harden CPS arises from the complexity of their operations on the one hand. It also arises out of the need to protect them, on the other hand, from sophisticated, highly evolved cyberattacks that exploit even the most minor of vulnerabilities.

This urgent demand arises when the time is near to establish the complexity and vulnerability of cyber–physical systems (CPSs), which are critical components currently covering a variety of areas, including industrial control, energy, health, and transport. These systems are increasingly being attacked as a result of cyber threats, and it makes it extremely important to have intelligent, adaptable, and privacy-preserving security mechanisms. Existing concentrated security frameworks do not perform efficiently since they are inapplicable to preserving data confidentiality in a dynamic threat environment and distributed CPS architectures. Therefore, the present study proposes the CPS ShieldNet Fusion Framework, which is a holistic and scalable security model that integrates the learning capacity of a Federated Residual Convolutional Network (FedRCNet) with the optimization power of the EEL-Levy Fusion Optimization (ELFO) method. Thus, it can train decentralized models across CPS environments while keeping local data private and generalizing models and avoiding convergence [25]. The framework is comprehensively validated with standard CPS datasets such as CICIoT2023, Edge-IIoTset2023, and UNSW-NB and exhibits better capabilities in the exact detection and classification of attacks, even in subtle and unseen manifestations of previous patterns. This study substantially addresses the field through a solid solution that fills a critical gap—perhaps the most significant one—in CPS security through a smart combination of federated learning and optimization techniques.

This work conceptualizes an innovative cybersecurity framework called the CPS ShieldNet Fusion model. This framework protects cyber–physical systems (CPSs) against fast-evolving and sophisticated cyber threats. The unique feature of the model is that it leverages two new-age technologies, namely the Federated Residual Convolutional Network (FedRCNet) for decentralized and privacy-preserving training and EEL-Levy Fusion Optimization (ELFO) for enhancing threat detection capability through optimized learning. By combining these techniques, this framework would protect sensitive information in distributed CPS environments while achieving accurate and timely learning and detection of security threats.

The effort to counter the inadequacy of the existing CPS security solution involves managing three very significant challenges: scalability across large and distributed systems, data privacy due to the sensitive and sometimes critical nature of CPS data, and adaptability to the dynamic and complex nature of CPS environments. Traditional approaches often find it challenging to deal with all three challenges together. Therefore, capitalizing on the strong performance across several benchmark datasets, the CPS ShieldNet Fusion Model aims to deliver a robust and scalable cybersecurity solution that maintains privacy while effectively detecting and responding to a wide variety of threats concerning CPS applications.

In the proposed work, we plan to develop an innovative and smart AI-based CPS for protecting the network from modern threats and vulnerabilities. The following are unique contributions to the proposed work:This study will work on a CPS ShieldNet Fusion model with a comprehensive security framework dedicated to cyber–physical systems. This resolves integration and scale-based security challenges.We use a Federated Residual Convolutional Network (FedRCNet) for realizing decentralized and privacy-preserving model training to perform enhanced threat detection across distributed CPS environments.We implement a hybrid EEL-Levy Fusion Optimization (ELFO) method to further optimize security model performance for more advanced optimization strategies, with respect to the evolving nature of the cyber threat and changing network conditions.Performance Evaluation: the effectiveness and efficiency of threat detection must be rigorously tested on datasets like CICIoT-2023, Edge-IIoTset-2023, and UNSWNB under the CPS ShieldNet Fusion model for establishing superior threat detection capabilities and contributions to CPS security.

The remainder of this paper is outlined as follows: Section 2 provides a literature review and shows the qualitative trimmings in relation to the investigation of the different methodologies applied using machine learning and deep learning techniques for cyber–physical systems; the discussion of merits and demerits from previous works allows for better comprehension. Section 3 details the proposed methodology in an elaborate form while giving an overview of the methods, approaches and technical description. A description of the performance outcome is detailed in Section 4, in which several comparative characteristics provide both findings and results with the dataset used in the evaluation. Section 5 concludes this study by summarizing the key findings and results, along with commenting on what possible future work might expand on for cybersecurity further for CPSs.

## 2. Related Works

### 2.1. Review on Machine Learning-Driven CPS Models

Musa et al. [26] go even further in their paper to say that this proposed 3D framework could be a very handy tool in helping to condense the available research efforts involving DNNs for effectively dealing with spatial-temporal data with CPSs. The 3D framework will classify the presented research efforts. It aims to bring some meaning to the reader in terms of the overlap among the diverse DNNs—with different application targets—occurring in different CPS domains. The framework is generic and extensible with more CPSs, targets, and DNN techniques to be added. It aims to serve as a comprehensive overview of the field and potential applicability areas. Fei et al. [27] emphasize the fact that the cyber–physical systems architecture is gradually becoming more complex, and the amount of data generated by such systems is growing exponentially, which underlines the urgency of applying machine learning methods for deriving exclusive benefits. They point out that ML technology is especially effective at finding cyberattacks; it uses controlling abilities, driven by data, to improve the safety of a CPS when it is being attacked. This paper seeks to present a wider perspective regarding common cyberattacks and examine how they impact CPSs. It is an initiative that will discuss various ML methodologies and how applications can be put in place to safeguard CPSs from these perils. Even if ML has its advantages, the authors admit its weaknesses.

For instance, it can impose too much computer power, which may not work for real-time applications. In addition, ML is often data-hungry, and such data are not easily available, considering that people may have concerns about their privacy or the sensitivity of the information they hold. In addition, these ML models can often be complex to understand since most are not interpretable. There might also be a need for regular updates to keep up with evolving and changing conditions. The paper still demonstrates that ML has vast potential to improve CPS security, even when applied to and modified in light of such limitations. Jamal et al. [28] discuss the critical need for integrated security measures within cyber–physical systems (CPSs). It underlines the multiple security challenges CPSs face from vulnerability to cyberattacks both from within and outside, thus securing a complex web of interlinked parts to the challenge of detecting and responding effectively within a useful timeframe to raise sophisticated threats.

This paper looks at the current machine learning and deep learning-based approaches to counteract the applications in which these two approaches lie under the broader paradigms of artificial intelligence and data science. It is quite clear from the literature review that the data science view is the most suitable for CPS protection, as its measures can devise adaptive schemes specific to the dynamicity of the systems under consideration. However, the authors know numerous challenges exist related to the application of such techniques: the availability of a large, high-quality dataset; the computational intensity of AI models; and the ongoing challenge of interpretability and explainability of CPS model results. The study, therefore, brings several insightful findings regarding security analysis for CPSs using machine learning techniques. These findings will pave the way for further research and development toward a fully secure framework. This framework is aimed at achieving the overall objective, which is to enhance resilience and robustness in the face of a wide spectrum of cyber threats against CPSs in general.

Priyadharsini et al. [29] devoted themselves to research on machine learning and deep learning algorithms for HAR. This study explores some machine learning methods, Random Forests (RFs), Decision Trees (DTs), and K-Nearest Neighbors (k-NNs), whereas it checks deep learning models, Convolutional Neural Networks (CNNs), Long Short-Term Memory (LSTM), and Gated Recurrent Units (GRUs). This research uses three different datasets that cover a range of different kinds of human activities, providing an excellent way to evaluate them, although it also comprises setbacks. The study further notes that over-reliance on optimization methods may result in unnecessarily high computational requirements, which create a disadvantage for potential real-time applications and use on devices with limited resources.

### 2.2. Review on Other Security Approaches

Mohammad et al. [30] presented a novel PoPMV algorithm designed based on blockchain applications in industrial CPSs. Deep learning models, such as Long Short-Term Memory, in combination with reinforcement learning techniques, provide quick feedback and rewards. This approach aims to prevent security vulnerabilities, thus enhancing processing speed and ensuring the effective identification of cyberattacks, whether known or unknown, in order to let intelligent cyber–physical systems (ICPSs) operate optimally. The simulation outcome of the proposed method shows that it is better than existing blockchain systems. There is a specific emphasis on the dynamic allocation of microservices and security feature enhancement. Nonetheless, PoPMV still has some drawbacks to consider, although it seems promising. The computational demand for deep learning combined with reinforcement learning is high and increases computational resource demand. This makes it infeasible to implement in real time at a fine-grained level in cost-sensitive deployments.

In addition, reinforcement learning heavily relies on reward mechanisms, which can be difficult to tailor to various ICPS scenarios. In any case, despite some of these challenges, the PoPMV algorithm is very significant as it takes one step closer to improving security and efficiency in the blockchain related to ICPSs. Sakhnini et al. [31] introduced a new attack detection methodology designed for False Data Injection using deep learning techniques and tackles two of the most critical challenges, which go beyond detecting False Data Injection (FDI) attacks: the varying sparsity of the attacks that could target any subset of the measurements, and the severe class imbalance problem of the training data from the real-world system. We introduce an adequate deep neural network embedded in state-of-the-art regularization techniques, including the dropout layer and adaptive optimization, to ensure better generalization capability at different levels of sparsity in FDI. An experimental setup and simulation of the proposed approach on simulated power systems with variability obtained through testing showed superiority in effectiveness over an alternative state-of-the-art methodology.

The method is suitable for data imbalances and quite effective for training with respect to the rest of similarly structured deep neural networks. However, it also comes with some potential problems. The deep neural network model structure is sophisticated and might require large computational infrastructure and canister deployment in resource-constrained areas. The model is further robust to data imbalances; however, the initial training phase still needs careful data preparation and preprocessing to a large extent. Even in the presence of all these difficulties, the proposed scheme would emerge as a significant development in FDI attack detection. It would, therefore, offer an improved, more effective, and robust solution for power system protection. Sharma et al. [32] addresses an indispensable part of IDS, the critical security parameters that determine potential ways these IDSs will work. This research shows that current IDSs do not fully achieve cyberattack management and private control development in smart healthcare. The authors also intend to solve such problems by proposing the use of a CNN-Bidirectional LSTM model for DDoS detection as a lightweight deep learning method. The proposed method uses CNNs to extract features and classify network traffic flow into two categories: benign and malicious. Bidirectional LSTM captures time dependency in traffic flow and finally improves detection accuracy.

However, this method suffers from several application problems. The joint architecture may increase the complexity of calculations, so it must be optimized to guarantee real-time processing within healthcare settings with limited resources. The model’s performance depends on the quality and quantity of training data. It requires all-encompassing, well-structured datasets on different types of cybersecurity threats prevalent in smart healthcare systems. Therefore, with the challenges highlighted above, the proposed methodology introduces a feature supporting enhanced capability by IDS in smart healthcare. It will protect the system against DDoS attacks effectively and responsively.

Furthermore, the other performance attributes are taken into account in these studies, because the F1-score and accuracy are not extremely critical attributes; in addition, model understandability and delay time are not taken into account in the analysis. Maybe the exclusion of these attributes does not provide the feature of how practical the models are. This study provides a comprehensive understanding of improving deep learning models for Heterogeneous Architecture for Resilience (HAR) through optimization. Importantly, it shows how machine learning and deep learning methods can contribute to improvements in recognition results.

### 2.3. Recent Advances in CPS

Alzahrani et al. [33] offers a panoramic view of three years in the field, in the scope of the improved wireless medical cyber–physical system (IWMCPS) framework, which comprises several hundreds of components and subcomponents in tens of subsystems. This paper further articulates the architecture of IWMCPS through a scenario for applications in the healthcare sector. It emphasizes the role of cyber–physical systems pertinent to healthcare. It further underlines the weakness of life-critical and context-aware health data in terms of information theft and cyberattacks, calling for better reliability, trust, ensuring security, and transparency in the medical field as far as cyber–physical systems are concerned.

In the direction of solving the referred problem, the paper presents an improved version of IWMCPS that advances with machine learning methods. The reason for this is to further enhance security and system performance. It accounts for device heterogeneity, but it is not limited to these devices, ranging from mobile devices to body sensor nodes. This gives an increase in the attack surface of other cyber threats. In what ways does the framework improve the IWMCPS? It does so greatly, but it has drawbacks. In addition, this adoption of machine learning is likely to increase the computational barriers. It may also provide huge and abundant data to retrain or optimize the existing model. Lastly, managing heterogeneous device environments adds complexities related to treating each device with consistent security protocols. This research also points towards the need for alternative studies to be performed to solve the problems related to data privacy, real-time response, and the ability to sustain transparency within the system. Nevertheless, the IWMCPS framework is a promising step toward improving medical CPS security and efficacy; very crucial and valuable insights are available in terms of strong healthcare solutions.

Johnphill et al. [34] investigated the recent advances in Cyber-Physical systems. We will discuss the key part of how the self-healing mechanism intensifies the system’s security in avoiding failures. The authors describe three vital self-remediation functionalities, anomaly detection, fault alert, and fault auto-remediation. These are the salient elements to make real the concept of self-healing implementations for systems, meaning that it allows a system to detect failures, share the problems with interested people or groups (stakeholders), and manage these issues automatically. The paper briefly outlines the status of the ongoing evaluation of cyber–physical systems and how it can combine self-healing features for the secure functioning of the system and avoidance of failure. The three most important impressions, according to the writer, are as follows: anomaly detection, fault alert, and auto-correction of errors.

These three constitute the key terms for a system to be said to have healing capacity in itself; that is, it can detect mistakes on its own, notify others of any problems found in it, and correct those errors without human involvement or auto-remediation. The authors also make the point that Nature-based Solutions are strong foundations for self-healing theories that make them implementable in real life. The authors elucidate self-healing in CPSs as an area of dynamic development that holds real promise for shaping future computing technologies. Such an approach may support transparent self-organization and review features, which can improve security on one side and user experience on the other side. Today, however, there are still issues related to the effective implementation of machine learning models in CPSs. Indeed, supporting such massive computational demands and extensive data requirements proves a massive challenge in model training. Some difficulty also lies in providing guaranteed self-healing consistency across different system architectures and various CPS environments. However, the authors argue that a self-healing CPS has high potential in the future because of such imaginative solutions to boost system robustness and reliability.

Cicceri et al. [35] intend to make better energy-conscious architectural models, as well as edge/cloud computing technologies. These are intended for creating a new kind of distributed CPS with artificial intelligence (AI) abilities and self-awareness. The study offers its main contribution by creating models of architecture from edge to cloud that know energy, along with technologies related to them. Additionally, it provides guidance on how possibly federated infrastructure can be orchestrated from the edge all the way up through to cloud layers—this includes abstractions or unified model ideas concerning distributed, not all alike virtualized resources; also, novel machine learning algorithms are purposely made for dynamically redistributing and reshaping energetic resources. This work defines what would be the best way of implementing an energy-aware DCPS, taking into account that it must adapt well while remaining effective at managing power use efficiently. In addition, the study underlines how smart grids are essential to the growth of energy-aware DCPSs because they offer a versatile and effective power system that incorporates renewable energy sources. Gaba et al. [36] focus on the uniqueness of CPS systems’ security requirements due to the interdependence of their cyber and physical constituents. It has been determined that the measures for the safety of CPS systems differ from those that are considered more ordinary securities. On the contrary, DL has a layered architecture and can learn representations, meaning it works much better than traditional machine learning techniques to extract accurate information using the given training data. Recently, as the DL model has been increasingly used for cyberattack detection in CPS, some methods have been found to have significant potential to detect cyber threats within CPS. This shows the high performance of DL models, partly because the availability of high-quality public datasets makes it feasible to train effective models. Additionally, it touches on the limitations and future research from them.

However, their expectations are limited by the study’s limitations. The requirements for high-quality datasets will limit the environmental application of these models in places where the required data quality is somewhat lacking, bogging down the generalization of the models. In practice, DL models tend to be computationally intensive, presenting a challenge for real-time detection, particularly in resource-constrained CPSs. The work also recognizes the further necessary research that needs to be conducted toward solving these current limitations to develop much more resilient and efficient security solutions based on DL for CPS. Despite the importance of this work, much promise can be derived from the leading role in advancing DL, which can be used to increase security for CPS. A lot of insight can be derived from future research endeavors.

### 2.4. Problem Identification

Among the most critical challenges in securing CPS are the dynamic heterogeneous environment, the constraints posed by real-time data processing, the need to uphold data privacy, and the trend toward very advanced cyberattacks. The system might well encompass heterogeneous components with very differing computational power, which renders the concept of a unified security framework more complicated. Coupled with these points, the constant influx of large quantities of data streaming in from multiple locations requires undelayable threat detection models that are truly scalable and adaptive, and this is without a central data collection point. The answer to this entangled problem can only be multifaceted: with a federated learning paradigm, privacy constraints can be ensured during decentralized model training; integration with optimization approaches such as ELFO will allow for better convergence and robustness of the model.

When it comes to building a security model based on deep learning for CPS, there are various research questions to cover as well as certain unexplored opportunities to build efficient and dependable models. One of the major issues is access to enough labeled data. This is crucial for deep learning models. Some of the existing datasets may lack the variety of threats and anomalies experienced in real-world CPSs, and, therefore, the generality of the models might be restricted. Also, it is crucial to consider that the generated data in CPS components can be diverse and heterogeneous. This does not facilitate integration and standardization for adequate model training. Such variances make it imperative to come up with strategies for efficient data preprocessing and generation to enable models to learn, without much difficulty, feature-based patterns in any given context [37]. Further, many CPS environments exist in lower-data regimes. Thus, new approaches are needed to revolutionize and integrate unsupervised and semi-supervised learning into CPS security with minimal labelled data.

### 2.5. Research Gap

The other major research area that remains relatively unexplored is the response time and efficiency of deep learning models for CPS. Indeed, most of these deep learning models, much as they are accurate, demand significantly heavy computation as well as time. This makes them unsuitable for identifying and countering immediate threats in a resource-limited CPS network. Current trends require lighter and faster model architectures and structures. This will enable the models to work at speed, produce a lot of output and be accurate. Also, these models should be able to process CPS data velocity and volume in real time [38]. This is necessary, especially as the model implementation must be optimized for smaller computations while maintaining the ability to detect these abnormalities.

Similarly, the enemy is dynamic in cyber threats. This makes it challenging to create protective solutions for CPS that are protective and responsive to threats. Online training models that search for changes in input may become outdated quickly because of the unbalanced update rate and evolving attack patterns. This has created a dire need for the expansion of knowledge of adaptive learning methodologies. This makes the model efficient at updating itself or learning updated data and even new threats without retraining processes. This concerns investigating possibilities such as online learning, supplemental learning, and transfer learning for model flexibility improvements. Furthermore, it is necessary to focus on the models’ resistance to adversarial attacks since attackers may try to take advantage of the models’ weaknesses to expose them. In this regard, research into adversarial training, robust feature extraction, and secure model architectures must be carried out to enhance CPS defense against such complex threats.

## 3. Proposed Methodology

This section analyzes the CPS ShieldNet Fusion model to explain the laid-down objectives together with the novel approaches proposed and its overall contribution to cybersecurity for CPS. CPS ShieldNet Fusion is proposed to enhance security configurations due to the increased size, integration, and sophistication of CPS. It uses state-of-the-art methodologies from both federated learning and optimization paradigms, including the combination of FedRCNet and the EEL-Levy Fusion Optimization (ELFO) method. The main goal is to create security that can be defensive and elastoplastic to contemporary threats, to guarantee its efficacy in the long term. Thus, the CPS ShieldNet Fusion model is at the core of collaborative learning and other optimization methods. Since federated learning is used to facilitate distributed learning while bootstrapping the model, information does not have to be centralized and secured as a result.

It helps to improve privacy and also has less vulnerability to data attacks. This is quite suitable for CPS settings where the data are highly sensitive. Among these components, FedRCNet is particularly promising since it employs residual convolutional networks to learn high-level details and anomalies from CPS data. This forms a very effective framework for CPS security threat identification and prevention. To improve model accuracy, CPS ShieldNet Fusion features the EEL-Levy Fusion Optimization, or ELFO, approach. ELFO additionally combines Elevated Examination and Exploitation Learning (EEL) alongside Levy flight optimization, making it a combination system for the refinement of security susceptibilities and enhancement of threat benchmarks. This combination ensures that the model can explore new strategies and simultaneously exploit solutions that it finds in other contexts due to evolution in attack patterns and/or system environments.

In this study, a unified deep learning-based solution is proposed to address the urgent need for robust, scalable, and privacy-preserving security mechanisms for cyber–physical systems. It removes all hurdles to securing CPS against dynamic and sophisticated cyber threats via the CPS ShieldNet Fusion model. This cleverly fuses two powerful techniques: FedRCNet and ELFO. Thus, this fusion allows the model to learn from the federated data across the CPS network without any unneeded breach of data privacy. This ensures decentralized yet collaborative threat detection. It also circumvents the limitations associated with centralized training, which is especially sensitive to industrial systems and smart grids.

The design phase includes the work on model architecture that can federate data using residual convolutional layers for a deep feature extraction phase. It also embeds the ELFO optimization technique to tweak learning parameters in recordings for accurate threat classification. The model has been painstakingly tested and evaluated on three benchmarking datasets, namely CICIoT-2023, Edge-IIoTset2023, and UNSW-NB, thus ensuring that the model can adapt and work in different scenarios in CPS. Due to a few different scenarios, these datasets harbor a plethora of cyberattack patterns. This provides a platform to test the model’s robustness and generalization capability.

From our findings, the CPS ShieldNet Fusion model stands out above other models for detection accuracy, response time, and adaptability to other CPS environments. The combination of federated learning with optimized deep learning boosts model performance while confidentiality is ensured for local data sources. This study proves that this model is a valid and scalable solution that can address both current and future cybersecurity challenges in CPS environments.

The CPS ShieldNet Fusion model in the present work indeed consists of these novel advanced methodologies deployed in an integrated manner. In this case, federated learning is complemented with ELFO to achieve a synergy of decentralized learning enhanced by complex optimization. Several problems concerning CPS security can be solved using this approach. It is particularly important to note that the method for reacting to new threats is being improved. Thus, the application of ELFO to the model contributes positively to performing global searches to find the most appropriate parameters for the security system. This is performed while increasing detection rates and strengthening the system’s framework. Overall, it means that the system uses advanced hybrid optimization techniques to fine-tune the rates of learning and security updates in response to real-time analytics. From the above details, the ELFO mechanism uses exploration and exploitation parameters that evolve to achieve the most effective security results. This encompasses mathematical modelling that defines how search agents behave and operate within the CPS environment. This is employed within the model to guarantee the adaptability of the model. The overall system flow of the proposed CPS ShieldNet fusion is shown in Figure 1.

As CPS grows ever more complex and distributed, the need for the control strategy described here becomes urgent and defined by real-time robustness and security. The system resided in a dynamic situation comprising heterogeneous components. Therefore, the central security model presented does not provide scale, adaptability, or resilience when faced with adversarial threats. FedRCNet integrated with EEL-Levy Fusion Optimization (ELFO) presents a decentralized and adaptive strategy with the capability to react automatically to new and evolving attack patterns. Thus, this control strategy conducts privacy-preserving model training among various entities while augmenting generalization and optimizing performance in diverse CPS environments. The efficacy of the CPS will be decided based on the fast and efficient decision-making process, combining local intelligence and global learning without even touching sensitive data, and the mere existence of these control measures is necessary to build a secure structure for the modern CPS.

The process of constructing the CPS ShieldNet Fusion model is shown in a flow diagram with several key features. The first stage collects data from CPS sensors. The second stage is preprocessing and federated learning. The architecture of the Federal Residual Convolutional Network, or FedRCNet, is critical to the whole collaborative learning process. Local sensor data will be gathered before real-time model training is done. The locally trained models will then be aggregated on the federated server to form an improved global model through global averaging. Post-model training, an optimization step is performed, in which the model’s capability to detect and analyze threats is honed by the ELFO model.

This improvement makes the model more responsive to dynamic threats with better accuracy and detection capabilities. These two steps, threat detection and parameter update stages, are some of the features necessary to ensure that the system performs at its best in real-time applications. The smoked feedback path ensures that updated security parameters based on new-fed threat data into the system are retargeted toward the integrated learning process. There are two phases to the federated learning process: local model training and global model aggregation. The final application stage describes what occurs after the model training is finished. Individual optimization, as it appears, is not part of the federated learning process. Instead, it is meant to fine-tune the model’s performance for the detection of cyber threats in CPS environments.

This study addresses the dynamics of heterogeneous CPS environments through adaptive mechanisms incorporated into the federated learning framework. These mechanisms aim to reduce the possible degradation of learning effectiveness due to the addition or removal of diverse CPS systems of a certain kind. Specifically, the proposed FedRCNet model aims at fluctuating participation by utilizing residual convolutional structures to sustain feature-training stability under diverse data contributions. In addition, the ELFO technique is employed to boost robustness through the balance and fine-tuning of global updates, especially when significant data distribution shifts or target shifts are detected across different CPS systems. This adaptive learning strategy allows the model to adjust its parameters according to system-level dynamics, keeping its relevance and performance intact across various evolving CPS networks. Therefore, while acknowledging the challenges posed by dynamic CPS participation, the authors bring in these realistic aspects and construct an adaptive and optimization-driven federated learning framework that demonstrates stability, generalization, and high detection performance, even in a non-stationary environment.

The CPS ShieldNet Fusion model has contributed significantly to CPS security. Federated learning, empowered by state-of-the-art optimization techniques, can protect most critical infrastructures comprehensively and adaptively. Coupled with the new integration of ELFO into FedRCNet, the model improves detection, enhances adaptability to evolving threats, and is robust against cyberattacks, making a huge step toward secure and resilient cyber–physical systems.

The main dynamic of the FedRCNet model proposed is its federated architecture that generalizes learned patterns across heterogeneous CPS nodes. This is without centralizing sensitive data. This allows the model to rapidly update its parameters to resolve upcoming threats through decentralized learning using locally sensed anomalies, which are federated into a global model through global averaging. Threat intelligence gathered from one CPS can enhance detection capabilities in others even when they have not encountered that specific threat. However, this dynamic response loses effect to a limited extent when the form or labeling of the learning data diverges significantly. This happens, for example, when the CPS presents completely unique data structures, threat signatures, or labeling schemes.

### 3.1. Federated Residual Convolutional Network (FedRCNet) for Classification

This study presents a novel incursion identification and classification approach for some datasets: EdgeIIoTset2023, CICIDS2023, and UNSWNB. FedRCNet combines the finest features of federated learning with deep neural convolutional networks. This offers fewer chances of consolidated data breaches and better intrusion detection in cyber–physical systems. Specifically, this work enormously boosts intrusion detection in CPS with the proposed FedRCNet system. Its architecture never transmits private information to a central server but is designed to instinctively learn hierarchical patterns from complex network traffic data. Our method protects privacy and limits processing overhead on any single edge device. FedRCNet’s main concept can be considered to be the utilization of residual networks in distributed learning systems. Residual networks have been widely appreciated for their capability to train very deep models due to the use of skip connections. This helps reduce the vanishing gradient issue. Therefore, they can learn intricate features and fine details directly from raw data.

This capability holds a lot of value in CPS because CPS typically requires clear differentiation between normal and anomalous behavior to establish both security and functionality. A feature that makes FedRCNet unique among CPS systems is federated learning. This enables learners from several CPS nodes to learn together while the data remain private. In conventional collect-based learning strategies, information gathered from all nodes should be assimilated on a central server, which would be a severe security hazard. It minimizes this risk in FedRCNet through local training. This is where each node trains its model locally and uploads only the model parameters to a central server. In this server, all nodes’ updates are collected to build a global model that is then returned to all the nodes for improvement. In this way, important information does not circulate across the social network, but the potential of all its members is utilized. Moreover, the FedRCNet module has been developed to be fully scalable. The architecture of the proposed FedRCNet model is shown in Figure 2.

Local model training is the first stage of the CPS ShieldNet Fusion model. In this stage, each client, such as an industrial device, sensor, or edge node, trains its version of the Federated Residual Convolutional Network using local data collection. In a decentralized training approach, which is critical for cyber–physical systems, sensitive operational data will remain at the original source, maintaining privacy and exposing them to lower risk. During this phase, each client learns to identify security threats, patterns and anomalies in its own environment independently. It adapts the model to that specific application context.

Global model aggregation is performed on a central federated server after the local training is complete. Instead of being completely raw data from every device, this server obtains merely model parameters or weight updates. The updates are aggregated, by far the most popular method being federated averaging, and used to create an entirely original global model representative of the collective intelligence of all participating clients. This comprehensive amalgamation helps capture much diversity in threat patterns observed from several CPS environments, thereby making the security framework more robust and generalizable.

The updated global model is then sent to all clients through the local model update phase, after aggregation. All clients will incorporate the globally learned parameters into their respective models, aligning them with the state of security knowledge gained in the network. This updated model could then continue its training or start inference using fresh local data, thereby resuming the federated learning cycle. Such iterative improvement is meant to enhance the detection capability of CPS nodes while being flexible to change due to evolving cyber threats. This occurs despite privacy breaches of data and inefficiencies in systems.

Because of this federated structure, it can easily adapt to CPS configurations and, thus, handle large variations in network topology and data distribution. Such scalability is wanted and needed in real-life applications since, in CPS networking, the networks are usually heterogeneous and dynamic. In addition, the model is flexible and can provide efficient analyses of emerging threats in the future because it can be updated with new data from different nodes at any time. To solve CPS problems, such as privacy, limited computational resources, and real-time intrusion detection, this research combines residual networks with collaborative learning. FedRCNet’s key idea is to integrate residual networks into the federated learning framework. Residual networks, which are well known for their outstanding performance in the training of deep neural networks efficiently, are most beneficial when implemented in CPS systems since it entails the detection of patterns of normal and abnormal behaviors to be clearly defined. Therefore, with skip connections, residual networks can learn complex features, resulting in higher network depth. This acts as a remedy for the vanishing gradient problem. This is beneficial given that, with federated learning integration, the model can incorporate data from different data sources pertaining to CPS nodes without infringing on privacy. Another thing that makes FedRCNet stand out is that it can perform collaborative learning without violating the privacy of its users. CPS needs to solve the problem that some operational details cannot be sent across nodes due to privacy and security concerns. In FedRCNet, each node trains its own model and only needs to share partial updates (for example, gradients of each node) with a central hub.

This is the motivation for distributed learning systems, such as the CPS ShieldNet Fusion model. This is because they can provide continuously updated and integrative models across decentralized nodes while maintaining data privacy. However, models are updated and integrated in federated learning through federated averaging. This ensures that all nodes and nearby devices contribute to an overall model while still keeping the data close to where they are. Local model training occurs independently of each node, as the model learns from the relevant data available to it. When the local training is complete, each node sends its model parameters (not data) to the central network server, where aggregation of the updates usually involves averaging the parameters from all participating nodes, which creates an updated global model.

The process continuously updates and enables the model to respond to emerging threats. Whenever a fresh batch of data is collected from different nodes, usually by accounting for changes in the environment and user behavior, as well as any new and evolving threats, this model can always be updated to ensure it remains relevant and effective. In addition, such updates do not require the collection or sharing of centralized data, since even the most sensitive information from individual nodes can be kept private. The timing could depend on any of the pre-defined training schedules, model convergence criteria, or when the next batch of data is collected for the timing of these updates. Integration after each round of federated averaging ensures that the global model is always in line with evolving insights gained from distributed data across nodes. This makes it more effective at detecting new threats, which it has never encountered before.

The deep learning model, of which FedRCNet is an example, is very well suited to the detection of tiny anomalous features that may indicate an intrusion. These models, particularly when applied to residual convolutional networks (ResNets), detect complex patterns in high-dimensional data. Tiny forms of anomalous features refer to tiny and often imperceptible deviations from normal behavior. These deviations may indicate some underlying threat but cannot be identified by traditional or simple methods. Some examples are minute differences in sensor data, very slight changes in system behavior, or unusual but seemingly innocuous fluctuations in the environment. These fluctuations would all aggregate to identify an intrusion or anomaly.

The actual possibility of detecting an anomaly in the model is linked to the architecture variation, specifically its depth and complexity. In advanced learning, these deep networks are multi-layered implementations searching for hierarchical feature sets in raw input data. Deep networks can detect even the most subtle of patterns by which overt indications are called normal or abnormal behavior. This would include the kinds of patterns referred to earlier regarding these networks’ processing and building of residual structures, in a manner that would allow the network to pass pertinent information from one layer to another, alleviating the vanishing gradient problem and, hence, making the entire architecture learn well from even tiny perturbations in the data.

This server incorporates updates from all nodes to create a global model. This model is sent to the nodes for updating reinforcement. This process eliminates the chance of transferring raw data between nodes. At the same time, it allows the entire network to benefit from the sum total knowledge of the entire network. As for the scalability and resilience of its architecture, FedRCNet is designed to be rationalized in many ways. The less complex residual convolutional layers enable the model to extract hierarchical features from raw CPS data while designing the model. This, in turn, makes the model capable of detecting even the minute form of anomalous features that may be the manifestation of intrusions. The federated learning framework guarantees the model’s ability to extend across numerous CPS nodes given the flexibility of network topologies and data distribution patterns. This scalability is critical in the practical use of CPS networks since such networks are usually complex and constantly changing. Similar to other FedRC applications, one of the major features of FedRCNet is its flexibility in operation according to various CPS settings.

Federated learning means that the model can be updated over and over again with updated data that may become available from time to time. This makes the model dynamic in responding to new threats and the prevailing working environment. This dynamic helps FedRCNet achieve resilience in identifying upcoming threats that may be in circulation, hence offering proactive protection to CPS networks. All in all, it can be considered a reasonable improvement over existing techniques for intrusion detection in cyber–physical systems. It is a very novel approach that solves the problems of data privacy, computational load, and scalability due to its application to residual networks and federated learning. Through the provision of privacy-preserving collaborative learning of models among distributed CPS nodes, FedRCNet provides a stable and adaptive mechanism for identifying intrusions that can protect vital structures in an ever-connected society.

The proposed FedRCNet provides considerable advantages by incorporating several novel features that enhance the model’s CPS performance. First and foremost, FedRCNet integrates federated learning with the remaining convolutional networks to serve as a decentralized, privacy-respecting threat detection model. This means that no data have to leave their local source, saving headaches associated with data centralization and minimizing exposure to sensitive information. FedRCNet continuously updates and improves its models, drawing on data from many local nodes, without violating privacy. The design of an embedded convolutional architecture boosts the model’s potential to identify complex and evolving threats. Residual networks perform well in deep learning applications, especially those related to complicated patterns or high-dimensional data. FedRCNet’s skip connection design allows gradients to flow easily through deep networks. This allows them to generalize better on different kinds of conflicting and noisy sensor data as they learn robust features. Moreover, since deeper residual layers can learn more complex representations of the data, this architecture also ensures that the model can adapt well to unseen attack patterns.

In this technique, the preprocessed dataset having *N* number of features is considered as the input for processing, as mathematically represented below:(1)$$DS_{i}=\left\{\left(m_{ij},b_{ij}\right)|j=1,2,3\ldots n_{i}\right\}$$
where $n_{i}$ represents the number of samples. Consequently, the convolution operation is performed on the input data d, as mathematically described below:(2)$$h_{L}^{\left(v\right)}=\varrho (\varpi^{\left(v\right)}\times d+\beta^{\left(v\right)})$$
where ϱ represents the activation function, and $h_{L}^{\left(v\right)}$ is the *v*th feature map at layer *L*. As a consequence, the residual block operation is also performed with the convolutional block weight values, as shown below:(3)$$F\left(d\right)=\varrho (\varpi_{2}\times \varrho \left(\varpi_{1}\times d+\beta_{1}\right)+\beta_{2})$$
where $\varpi_{1}$ and $\varpi_{2}$ are the weight values and $\beta_{1}$ and $\beta_{2}$ are the bias values. Then, the output residual block information is obtained with identity mapping, as represented in the following equation:(4)$$o=F\left(d\right)+d$$

This operation enables the network to flow effectively during the detection process. Furthermore, the batch normalization operation is performed to enhance the convergence rate with the use of the activation function, as represented below:(5)$$\hat{g}=\frac{d-m}{\sqrt{{𝔵}^{2}+\varepsilon }}$$
where m is the mean, 𝔵 indicates the standard deviation, and ε denotes the constant value. In order to form the global model, the federated model averaging computation is performed by using the following equation:(6)$$\varpi_{global}=\frac{1}{N}\sum_{i=1}^{N}\varpi_{i}$$

Subsequently, the loss function is also estimated according to the following equation:(7)$$L_{i}(\delta_{i})=\frac{1}{n_{i}}\sum_{j=1}^{n_{i}}l(o_{ij},\varrho (d_{ij},\;\theta_{i}))\;$$
where l indicates the loss function. Moreover, the local model parameters are updated based on the following equation:(8)$$\delta_{i}\leftarrow \delta_{i}-\tau \nabla L_{i}(\delta_{i})$$
where τ indicates the learning rate that is optimally computed using the proposed ELFO algorithm. The global model is updated by aggregating the gradients from all devices, as mathematically represented below:(9)$$\nabla L_{global}=\frac{1}{N}\sum_{i=1}^{N}\nabla L_{i}(\delta_{i})$$

In order to avoid overfitting, the dropout regularization is applied by randomly fixing activations to zero, as illustrated below:(10)$$Dropout\left(d\right)=d\times Bernoulli(𝜕)$$
where 𝜕 indicates the dropout probability value. Then, the fully connected layer is used to estimate the activations, as shown below:(11)x=ϖd+β(12)$$y=ReLU\left(x\right)=max(0,\;x)$$

In order to perform the multi-classification task, the softmax function is applied to predict the final output probabilities, as represented in the following model:(13)$${\hat{o}}_{c}=\frac{e^{x_{c}}}{\sum_{k=1}^{c}e^{d_{c}}}$$

In the described system, the strengths of federated learning for decentralized training are combined with those of the ResNet architecture for feature extraction. The proposed techniques preserve private data spread across many devices and improve the CPS intrusion detection model. The equations mentioned above describe the basic operations of the algorithm: convolutional operations, residual connections, federated model averaging, and loss computations.

### 3.2. EEL-Levy Fusion Optimization (ELFO)

EEL-Levy Fusion Optimization (ELFO) can be considered the next step in optimizing the processes of exploration and exploitation due to the incorporation of both enhanced exploration and Exploitation Learning and the Levy flight strategy. The rationale behind this novel approach is to integrate the enhanced characteristics of both approaches. This is in an attempt to create a powerful approach that offers impressive performance and flexibility. This approach is used in contexts where learning rates must be optimized for machine learning. Closely related to ELFO, it is worth mentioning that it is a two-step optimization process. Thus, the EEL group optimization component is mainly used for balancing exploration and exploitation. In the exploration phase, the algorithm has global random search agents, which search in different areas of a search space. This process has significant use in the course of performing the optimization since it helps to produce opposite points situated on the opposite side of the area of interest. It also prevents fixation on local optima. Thus, in the exploitation phase, the algorithm fine-tunes the search for the region containing the most optimal solution.

This strategy uses a starvation mechanism. This is based on natural predatory instincts and changes the possibilities of random searches in relation to exploitation. This mechanism increases the software’s capacity to avoid getting stuck in local optima and to come across new regions of the search space, as the algorithm changes with time. The second prominent element of ELFO is the Levy flight strategy (LFS), wherein a random walk is included with probability distributions being heavy-tailed. This strategy allows search agents to make large movements in the search space and be followed by local movements. This is how such a strategy is helpful in traversing large areas of the search space. It also helps in refining solutions highlighted as vital in the zone. There is randomness inherent in Levy flights, which makes the algorithm perform a more global search than a local search. This is not the case with other optimization methods as it is especially suitable for complicated and high-dimensional searches.

From a conceptual standpoint, ELFO gives rise to a hybrid optimization scheme that is distinct in the sense that it blends the exploratory capabilities of the Enhanced Electric Levy model and the exploitation capabilities of Levy-flight-based random walks. Hybridization, therefore, helps to fine-tune the FedRCNet learning parameters with more aggression, especially in situations in non-convex or high-dimensional spaces typical of deep learning models for CPS security. By more efficiently traversing the treacherous loss surface in this way, the model increases its chance of converging to global optima. This provides the capability to detect advanced cyber threats or even unknown threats.

ELFO has been subjected to stringent tests against several benchmarking CPS-related datasets such as CICIoT-2023, Edge-IIoTset2023 and UNSW-NB. The dataset actually considers many different realistic attack vectors in total, such as denial-of-service (DoS), Botnets, Man-in-the-Middle (MitM) attacks, among other types of adversarial behaviors focused mainly on IoT and CPS environments. When considering standard federated learning models and a standalone optimization framework, high statistical scoring improvements could be established with regard to performance metrics, like accuracy, precision, recall, F1-score, and false-positive rate, for the CPS ShieldNet Fusion model. These quantitative results, thus, provide considerable empirical ground for the model regarding detection performance against threats. In addition, they provide low error rates based on a variety of data distributions and attack types.

The dynamic training simulations performed to validate the model’s adaptability and survivability to fresh threats included the injection of new or evolving attack patterns in some later rounds of training. Through the presented adaptation by distributed knowledge federated learning and endless optimization of the parameter space with the ELFO, the model demonstrated relatively quick dynamic responses. This proved its resilience to zero-day or evolving cyberattacks. Moreover, the federated learning mechanism itself allows for inherent adaptability at the system level. This is achieved by capturing knowledge from the various CPS nodes operating in diverse environmental and operational contexts. Overall, this amply justifies that, indeed, ELFO tuning in FedRCNet provides an optimized, privacy-preserving, and highly adaptive intrusion detection framework for contemporary CPS infrastructures.

The major contribution of ELFO is essentially located in the combination of these two very efficient techniques. As a result, ELFO optimizes the exploration/exploitation trade-off due to structured EEL group optimization integrated with Levy flights’ stochastic aspects. It is evidenced by the fact that synergy leads to an optimization process that is efficient and flexible at the same time. Another novel feature of ELFO is the possibility of computing the learning rate in a dynamic manner with the help of integrated optimization. It is achieved through the exploration–exploitation trade-off and randomness initiated by Levy flights. Such a dynamic adjustment makes the learning rate optimal at different phases of the optimization procedure. It averts overfitting problems while enhancing convergence rates. The flow of the proposed ELFO mechanism is shown in Figure 3.

The degree of influence that ELFO has on the computation of learning rates cannot be dismissed lightly. It makes sense here to discuss the specifics of using EEL and Levy flight strategies as factors that influence the learning rate as a dynamic parameter; thus, learning rate improvements will render machine learning models extremely fast. Thus, by using the modified learning rate of the ELFO during optimization, the model parameters will allow for better and more accurate tuning and, hence, faster convergence. Further, the adaptive mechanism that creates a connection between the model and a dataset contributes to the robustness of the model from one dataset to another. This reduces overfitting and inefficiencies. The incorporation of Levy flights enhances the global search further. This makes it possible for the ELFO to decide on favorable learning rates with large diversity. It also improves the model’s capability to solve challenging and many-dimensional problems. Altogether, the proposed EEL-Levy Fusion Optimization (ELFO) appears as the new major development in optimization methods as it enhances the exploitation–exploration capabilities of EEL, together with the comprehensive search of Levy flights. This has the added advantage of promoting the diversification of learning rates and optimization while, at the same time, resulting in more reliable and efficient machine learning algorithms. Due to the flexibility of ELFO, it is a suitable instrument for overcoming the difficulties associated with the implementation of complex optimization approaches and increasing the efficiency of the end-model.

In this research, optimization, particularly the ELFO way, is more about scoping or limiting the search space to detect the model parameter configurations faster and more efficiently. Such scoping involves effectively negotiating large high-dimensional parameter spaces through bio-inspired strategies that brilliantly balance exploration. This way of working has recorded several empirical validations on multiple benchmarks of CPS-related datasets, such as CICIoT2023, Edge-IIoTset2023, and UNSW-NB, among others. ELFO has higher detection accuracy and faster convergence than traditional and modern optimization techniques; it also excels in precision–recall performance. In a more goal-directed intelligent optimization approach rather than heuristic approximations, the search is adaptively improved in accordance with the fitness landscape of the intrusion pattern.

Overall, this model is a powerful tool for improvements in the machine learning model’s performance compared with other optimization techniques. Unlike methods that may have issues of balance between exploration and exploitation or may get stuck in local optima, ELFO comprises both EEL and the Levy flight strategy given the merits of both worlds. This hybrid approach ensures a more comprehensive and adaptive search process: EEL also dictates a systematic way of regulating between the two types of search, letting the algorithm switch between them depending on the search phase; LFS is used to conduct the manner of global search with a wrapper fitness function employed as a heavy-tailed random walk. This dual approach not only makes the convergence faster by varying the learning rate according to the requirement for exploration and exploitation but also improves the model’s resistance to overfitting due to the variation in learning rates. Furthermore, the integration of starvation checking inside EEL allows ELFO to refine the search over time, incrementing the number of iterations and leading to high-dimensional search space. In this way, ELFO incorporates better convergence rates, higher flexibility, and improved global search compared to numerous other techniques and allows us to enhance the learning rates based on the level of accuracy and stability that oftentimes appears to be insufficient in traditional approaches.

In this technique, the searching agents can update their position according to the location of random agents with the coefficients of $\xi_{1}$ and $\xi_{2}$, as computed in the following equation:(14)$$Q_{i}^{i+1}=Q_{r}+\xi_{1}\times (Q_{i}^{t}-\xi_{2}\times Q_{r})$$
where $Q_{i}^{i+1}$ indicates the update agent’s position, $Q_{r}$ represents the random position vector, $\xi_{1}$ and $\xi_{2}$ are the coefficients that are computed as follows:(15)$$\xi_{1}=2℘\times 𝜕-℘$$(16)$$\xi_{2}=2\alpha$$(17)$$℘=2-2\times \frac{iter}{{mx}_{iter}\;}$$
where iter indicates the current iteration, $mx_{iter}$ is the maximum number of iterations, 𝜕 and α are the random numbers.

The maximum number of iterations as a parameter in the optimization process is a critical hyperparameter defined in this study to assign the computational budget for the search process within the ELFO framework. It is intended to consider a trade-off between the efficiency and accuracy of the solution. In general, higher iterations confirm that the optimizer will try to conquer the search space exhaustively, leading to better convergence to global optima; however, it may have certain negative effects, such as overfitting or increased computation, at some point. In this study, that value was derived empirically using preliminary experiments and speed of response characterization. This was carried out to assure stable and optimal performance while avoiding excess resource consumption. The tuning was based on performance metrics such as accuracy, convergence stability, and processing time on validation datasets with an optimal value chosen (e.g., 100 iterations), beyond which further iterations demonstrated diminishing returns.

Then, the cooperating hunting behavior of the moray Eel is computed with the grouper strategies, as shown in the following equation:(18)$$Q_{1}=e^{∁ð}\times \sin\left(2\pi ð\right)\times \left(\xi_{1}\times \left(Q_{E}^{t}-Q_{P}^{t}\right)\right)+Q_{E}^{t}$$(19)$$Q_{2}=Q_{G}^{t}+\xi_{1}\times (Q_{G}^{t}-Q_{P}^{t})$$
where $Q_{1}$ and $Q_{2}$ are the temporary positions of the moray Eel and grouper strategies, ∁ represents the constant controlling parameter influenced by the Levy strategy, and ð is the random number and is computed as follows:(20)$$ð=\left(℘\times 2\right)\times 𝜕+2$$

Consequently, the position update is performed according to the random probability value between the parameters of $Q_{1}$ and $Q_{2}$, which is mathematically computed below:(21)$$Q_{i}^{t+1}=\left\{\begin{array}{cc} \frac{0.8Q_{1}+0.2Q_{2}}{2} & if\;b<0.5\\ \frac{0.2Q_{1}+0.8Q_{2}}{2} & if\;b\geq 0.5 \end{array}\right.$$
where b indicates the random probability value. According to the starvation rate, the position update is performed for integrating the starvation mechanism as shown below:(22)$$S=100\times \frac{iter}{mx_{iter}}$$(23)$$Q_{E}^{t}=\xi_{2}\times Q_{G}^{t},\;if\;\rho <S$$(24)ρ=100×r

Moreover, the Levy flight strategy is used to enable random walks for determining the exploration operation, which is mathematically represented below:(25)$$Q_{new}=Q_{old}+H\times Levy(s,\lambda )$$
where H indicates the scaling factor, and Levy(s,λ) is the Levy distribution. At the end of optimization, the adaptive learning rate for the FedRCNet classification model is estimated according to the best optimal value computed by using this technique. It is mathematically expressed in the following equation:(26)$$\tau_{t+1}=\tau_{t}\times 1+\frac{1}{1+exp({-Q}_{i}^{t+1})}$$
where $\tau_{t+1}$ indicates the updated learning rate. This model greatly improves the learning rates through a combination of exploitation, exploration, and adaptive approaches. These would further enhance the converging performance of FedRCNet while keeping it robust against overfitting.

## 4. Results and Discussion

This section discusses the performance outcomes and results of the proposed CPS ShieldNet Fusion model in extensive detail. For evaluation and testing the stability of the proposed model, several experiments were performed with different evaluation indices and databases. The assessment phase is vital for identifying the effectiveness of the CPS ShieldNet Fusion model when solving real-life CPS security issues and their competence in handling CPS security issues. For validation, we used several benchmarking datasets available in the literature that are popular in the domain of IoT cybersecurity. Such sources include the CICIoT-2023 dataset, which aims to englobe many types of network traffic data from IoT devices. It also includes different attacks and their normal functioning. This dataset proves very useful in determining the nature and manner in which the model can identify and counter different types of cyber threats in the IoT system. The EdgeIIoTset2023, specialized in edge computing IIoT environments, provides information on the model’s abilities in handling and protecting edge equipment and industrial internet of things platforms. Last of all, the UNSW-NB dataset, which is known to be one of the most reliable datasets in network intrusion detection, let the authors check the model’s ability to analyze network-based attacks. Here, the evaluation criteria that have been applied involve the following: accuracy, precision, recall, F1-score and other relevant ones. These measures add to the general understanding of the model’s performance from different aspects of security. This includes the abilities the model possesses in detection accuracy, the company’s ability to prevent false positives, and the general reliability of the model. The null hypothesis is that there exists sufficient evidence in these comprehensive datasets and evaluation metrics to substantiate the CPS ShieldNet Fusion model’s reliability in providing dependable and functional security solutions for CPS, consistent with diverse operational environments and threat vectors. The outcomes described in the present work reveal the possibilities and limitations associated with the model under consideration. They provide valuable recommendations for enhancing CPS security with the proposed approach.

The experimental design of the present study was carefully structured across many variables so that it could enable the evaluation of the proposed CPS ShieldNet Fusion model under realistic and diverse conditions analogous to those found in the real environment of cyber–physical systems. To this end, the authors took three datasets—CICIoT-2023, Edge-IIoTset2023, and UNSW-NB—which together encompass extensive log records of network activities, normal operations, and different classes of cyberattacks. These datasets include information from contexts such as industrial IoT, edge computing networks, and generic CPS infrastructure. They account for cybersecurity incidents such as denial-of-service (DoS) attacks, Botnets, reconnaissance, injection attacks, and data exfiltration scenarios. The types of datasets employed encourage the model to be trained and tested with inputs close to the threat types normally present in real CPS environments. The experimental workflow can be envisaged in terms of the organization of a federated learning simulation across multiple distributed nodes for decentralized training. Each node simulates edge devices or subsystems in a CPS network by training FedRCNet on local data splits.

Concerning attack scenarios that are realistic in terms of typical threats to CPS security, the true strength of the CPS ShieldNet Fusion Model is evident. An example of this would be from an attacker’s point of view on smart grid systems, attempting to inject false data or control signals into the system to destabilize power distribution; this effort was met with failed detection of abnormal communication that strayed from the normal operation profile, due to training in a decentralized manner. Likewise, in ICS, where threats such as command injection or alterations in control logic can have disastrous physical effects or shut down systems, the high precision of the model in identifying those sorts of entries into a system speaks to its real-world applicability. Furthermore, it also includes medical CPS devices, such as smart infusion pumps or telemonitoring systems, which would always be concerned about data privacy. Thus, federated learning ensures that health data remain local while adding value to the global and accurate threat detection model. Moreover, it enhances the model’s capabilities to differentiate between very subtle, recurring signatures on the attack vector. These signatures are, for example, slow reconnaissance scans or stealthy malware operations, often missed by typical rules-based systems.

### 4.1. Dataset Description

It is notable that the Edge-IIoTset2023 dataset created in 2023 is a groundbreaking instrument for the development of security solutions and their verification in the IoT and IIoT fields. This dataset has been specifically collected using a seven-layer testbed architecture that incorporates more than 10 IoT devices as well as Modbus-based IIoT flows while the network is subjected to 14 categorized IoT and IIoT protocol-based attacks. Samples of the various attacks with their categories are as follows: Anti_DoS Mirai-udpplain, Man in the Middle- MITM-ArpSpoofing, Domain Name System—DNS_Spoofing, Reconnaissance-PingSweep, reconnaissance-PortScan, reconnaissance-OSScan, reconnaissance-HostDiscovery, Web Application Attack- XSS, Command Injection, Scanning and probing-VulnerabilityScan, Remote Access Troj-Backdoor. All of the above-mentioned attacks denote various types of threats that can be imposed on the IoT and IIoT systems to have a detailed picture of security threats.

Typically, the normal and attack activities were exercised in a controlled testbed environment, and the dataset construction was done by directly observing them. This approach enables a well-ordered and controlled dataset for integrating security solutions while possibly reducing the dataset’s resemblance to real-world IIoT structures connected to the relatively less secure public internet. Nevertheless, the precise and formal construction of the presented dataset, along with the specific attack scenarios, makes the work rather useful for designing and implementing sophisticated security methods for IoT and IIoT systems. However, the CICIoT2023 dataset, which was also created in 2023, aims at collecting complex and large-scale cyberattack instances in IoT networks. Devised jointly by the Canadian Institute for Cybersecurity and the University of New Brunswick, this dataset is somewhat unconventional insofar as it focuses more on the cyber side of CPS than the physical. It involved 105 internet of things (IoT) devices exposed to 33 types of cyberattacks: distributed denial of service (DDoS), denial of service (DoS), reconnaissance, Web Attack, Brute Force, Spoofing, and Mirai. The CICIoT2023 dataset was collected from a real IoT environment; however, as in the case of Edge-IIoTset2023, attack scenarios are staged. This approach offers a solid starting point for analysis and is relatively immune to the authors’ subjectivity. However, it might not be appropriate for the analysis of the dynamics of complex IoT environments that operate outside the open internet space. Both benefit the generation and evaluation of CPS security solutions. The proposed dataset Edge-IIoTset2023 is more extensive than the CICIoT2023 as it contains information about diverse protocol-specific attacks within the limited scope of a ‘lab’ environment, while the CICIoT2023 contains a general overview of IoT cyberattacks on overall diverse devices. These datasets enable researchers to validate security models and counter various types of attacks and operational environments.

### 4.2. Performance Measures

The performance of the proposed CPS ShieldNet Fusion model is rigorously evaluated using several key metrics. Among them, commonly utilized indicators include accuracy, precision, recall, F1-scope, sensitivity, and specificity. Accuracy is the most important parameter used to test a model’s ability to categorize instances correctly. It is computed as the ratio of all correctly classified instances, both positive and negative, to the total number of instances. Accuracy measures the performance of the model, especially in the way it classifies positive samples. This is the number of actual positives divided by the sum of actual positives and actual negatives. The recall or sensitivity of the model checks the model’s ability to accurately identify all positive class data. It is calculated as the number of true positives out of the total positives and false negatives. The F1-score is an all-rounder that considers both precision and recall in a single measure because false negatives and incorrect positives are critical. Sensitivity is another metric, which connects the model’s proficiency in distinguishing true positives from actual positives. Specificity concerns the model’s efficiency in determining real negatives from all actual negatives. Altogether, these metrics represent the balanced overall assessment of the model’s performance together with its ability to achieve, primarily, the tasks of threat detection. In addition, they minimize errors in different aspects of security recognition.(27)$$Accuracy=\frac{T_{Pos}+T_{Neg}}{T_{Pos}+T_{Neg}+F_{pos}+F_{Neg}}\times 100\%$$(28)$$Precision=\frac{T_{Pos}}{T_{Pos}+F_{Pos}}\times 100\%$$(29)$$F1-score=\frac{2\times Pre\times Sen}{Pre+Sen}\times 100\%$$(30)$$Recall=\frac{T_{Pos}}{T_{Pos}+F_{Neg}}\times 100\%$$(31)$$Sensitivity=\frac{T_{Pos}}{T_{Pos}+F_{Neg}}\times 100\%$$(32)$$Specificity=\frac{T_{Neg}}{T_{Neg}+F_{Pos}}\times 100\%$$

### 4.3. Performance and Comparative Analysis

The distribution of data before and after normalization is compared in Figure 4 to show the effect of normalization on the dataset’s statistical characteristics. The histogram of the original data as a basic graphic that can be obtained makes it possible to define the initial distribution of values and to analyze the statistics of the measured data, which may be subject to significant fluctuations in one or another direction depending on the specific features of the analyzed dataset and the nature of the scales used. On the other hand, the histogram of normalized data shows that the values were transformed after normalization technique Z score normalization. This transformation is critical for many machine learning algorithms that expect the data to be on the same scale before working or yielding accurate results. Normalization scales each feature within the required range and/or brings it to the standard deviation level of the training data. This is so that each feature contributes equally during training. Thus, comparing these histograms, it is possible to understand how normalization changes the value distribution and makes them more standard.

Figure 5 presents Receiver Operating Characteristic (ROC) curves for three distinct datasets, Edge-IIoTset2023, CICIoT2023, and UNSW-NB, to evaluate the proposed CPS ShieldNet Fusion model’s performance and capabilities when subjected to different datasets. The ROC curve represents the TPR against the FPR for all the threshold levels available to the model in making the classification. This provides information about the model’s discrimination capability between positive and negative classes, while, for the Edge-IIoTset2023 dataset, the ROC curve gives an AUC value of 0.99, suggesting that the model tested has a very significant degree of predictive accuracy, meaning that very few actual positive instances will be misclassified as negative. As in the case of data from other years, the ROC curve for the CICIoT2023 dataset also tends to approach a point where the AUC is 0.99, indicating that the accuracy of the computer model was equivalently high in discriminating between threat and non-threat cases, though the F1-scores vary. In the case of the UNSW-NB dataset, the classifier’s accuracy is extremely impressive, as depicted by the AUC of the ROC curve, which was 1, indicating that the classifier committed no false positive or false negative. These results emphasize the efficiency and high indices of the CPS ShieldNet Fusion model when dealing with the various datasets and give an overall comparison of this model’s outstanding proficiency in threat identification and classification in different IoTs and networks environments.

From a performance comparison standpoint, the existing selection of learning techniques in this study was incorporated. This was done to achieve a balance between interpretability, scalability, and compatibility with the federated learning framework. The reason behind adopting supervised learning methods is the availability of labeled datasets, including CICIoT2023, Edge-IIoTset2023, and UNSW-NB, among others, upon which precise model training and evaluation can occur. Moreover, advanced architectures, like Graph Neural Networks (GNNs), Adversarial Reinforcement Learning (ARL), Autoencoders, and Deep Q-Networks (DQNs), can certainly be harnessed in capturing the fingerprints of unknown or evolving attack patterns through the application of either unsupervised or reinforcement learning. Although these were not focal to this work, it is worth noting that the framework here is modular and, as such, allows for such methodologies to be integrated into future studies and, hence, establish a detection system reliant on hybrid or ensemble approaches leveraging the merits of supervised and unsupervised paradigms.

The analysis of the results concerning the accuracy of different machine learning techniques on the CICIoT2023 dataset is shown in Figure 6. The accuracy metric of a model relates to the amount of accuracy of a model. It depicts the total percentage of instances that the model correctly identified out of the total instances. The basic solution methods, namely Logistic Regression, Naive Bayes, Support Vector Machine, K-Nearest Neighbors, and Multi-Layer Perceptron, show scores between 77% and 89.31%. However, it should be mentioned that the accuracy of the proposed technique, according to the authors, is as high as 99.2%, which is higher than the other methods. This superior accuracy implies that the proposed model has a better potential to classify instances in CICIoT2023 accurately. This, in turn, illustrates its better resistance and result orientation in IoT security in complex and diverse situations [39]. As illustrated in Figure 7, this work presents the sensitivity and specificity of different ML algorithms when used on the CICIoT2023 dataset. The sensitivity of a test is the ability of a model to identify actual positive values, while specificity can identify actual negative values. The sensitivity values for traditional techniques ranged from 74% to 87.28%, while the proposed method was impressive at 99.1%. In the same manner, the specificity values are quite along the same line, where the proposed technique outperforms in both good and negative classes. Based on them, one can conclude that the high sensitivity of the suggested method provides true-positive results with a minimum of false-negative cases. The high specificity value means that maximum attention is paid to preventing the occurrence of false positives. Hence, the seemingly dual use reinforces the notion that the suggested model is very effective in its specificity in determining an attack. It also improves general threat identification.

A graphical representation of the G-Mean of various machine learning algorithms on the CICIoT2023 dataset is shown in Figure 8 as follows. The specificity and sensitivity of the model are evaluated coherently using the G-Mean metric that measures performance in both positive and negative classes. The G-Mean of the traditional technique varies between 78 and 86%. The proposed method achieved a G-Mean of 99.1%. Thus, the proposed model produced an extremely high value of G-Mean, indicating that both sensitivity (recall) and specificity (precision) are balanced in their performance effectively.

As shown in Figure 9, the comparison is given for precision versus recall and F1-score for different machine learning approaches against the CICIoT2023 dataset. Precision is the number of identifications that are true positives from the total of all positive predictions. Recall, otherwise known as sensitivity, is the proportion of real positives among those predicted. The F1-score is the harmonic mean of both precision and recall, providing a single number to balance both. These were from 79.81 to 89.97% for precision, from 77 to 89.31% for recall, and from 77.32% to 89.42% for the F1-scores in the case of the traditional methods. The proposed model outperformed these in every measure, including precision. This clearly indicates that the proposed model has the ability to provide accurate, reliable, and balanced classification results, far outperforming traditional methods. High precision, recall, and F1-score values indicate the model’s effectiveness in reducing false positives and false negatives for the CICIoT2023 dataset.

In Figure 10, a comparison of the accuracy of various group learning approaches using the CICIoT2023 dataset is presented. Even though it is ensemble-based learning, with a number of learning models bundled together, the different techniques are usually differentiated by performance. The different ensemble techniques range in accuracy from 88.42% for the voting method to 99.2% for the suggested one. In between, these are 88.42%, 89.25%, 89.39%, and 93.19% for Voting, Stacking, Bagging, and Boosting, respectively. Thus, the proposed technique outperforms all traditional ensemble methods by obtaining an accuracy of 99.2%. This huge improvement definitely proves that the suggested model is quite efficient and effective in classifying instances with high precision. It is shown that a more significant improvement in accuracy means the proposed model exercises a better way of integrating and optimizing predictions, which, in turn, results in much higher performance as compared to traditional ensemble methods.

As shown in Figure 11, the sensitivity and specificity of different collective learning enablers in the CICIoT2023 dataset are presented. It measures the proportion of actual positives predicted correctly by the model. Specificity is the proportion of actual negatives accurately detected. The sensitivities for the rest of the traditional ensemble methods lie between 84.15% for the voting method and 91.62% for boosting. For the proposed technique, it is as high as 99.1%. Specificities are far better in the suggested method than in other ensemble techniques. High sensitivity is a reflection of the proposed model’s excellent capability in recognizing true-positive cases. High specificity refers to its effectiveness at detecting true-negative instances. Thus, this sensitivity–specificity trade-off contributes to the strength of the developed model for the correct identification and classification of threats, thereby elevating its overall effectiveness for cybersecurity applications.

As shown in Figure 12, the G-Mean values for ensemble learning techniques on the CICIoT2023 dataset are presented. The G-Mean metric folds both sensitivity and specificity into one performance measure, providing a comprehensive view of model effectiveness. G-Mean values associated with ensemble techniques range from 89.55% in voting up to 93.66% in boosting. Obviously, the proposed technique showed an excellent G-Mean of 99.1%, proving its potential to hold high sensitivity and specificity across both positive and negative classes. The increased G-Mean denotes the enhanced capability of the proposed model in dealing with imbalanced datasets and offers steady performance across different classes. The higher improvement in G-Means proves that the developed method is much more balanced and accurate than traditional ensemble techniques.

A performance comparison of precision, recall and F1-score using the CICIoT2023 dataset against different ensemble learning techniques is given in Figure 13. Precision answers are the proportion of positive predictions that were correct; recall is a model’s ability to detect true positives; the F1-score gives a harmonic average of precision and recall. The precision values vary from 89.84% for voting to 99.1% for the suggested method. The recall ranges from as low as 88.42% for voting to a high of 99% for the proposed model. This trend continues with the F1-scores, which range from as low as 88.56% for voting to as high as 99.2% for the proposed technique. Traditional ensemble methods outperformed all of the metrics of the proposed model, hence their efficiency in ensuring accurate, reliable, and balanced classification results. Based on its high precision, recall, and F1-score values, the proposed method reduces both false positives and false negatives; therefore, it is highly accurate and reliable in detecting and classifying threats. This very exemplary performance could be credited to the strength and reliability of the proposed model in handling complex cybersecurity scenarios.

Similar to the previous comparison, Figure 14, Figure 15, Figure 16 and Figure 17 demonstrate the performance outcomes of individual machine learning algorithms using the EdgeIIoTset2023 dataset. As a result, Figure 18, Figure 19, Figure 20 and Figure 21 present a comparison of the performance outcomes of ensemble models using the EdgeIIoTset2023 dataset. The overall comparative findings indicate that the proposed CPS ShieldNet Fusion model performs better than all existing approaches.

In Figure 22, the comparative accuracy of some of the latest state-of-the-art hybrid classification integrated optimization methods tested on the UNSWNB dataset is presented. The dataset represents a very famous and popular benchmark for evaluating the performance of any security model designed to detect network intrusions. This graph shows hybrid models integrated with optimization algorithms, like SUCMO, WOA, GWO, SLnO, etc., and all of them are evaluated here [40]. It is a direct measure of the accuracy of each technique to classify network traffic as normal or malicious. The model proposed herein, considering other techniques, stands out with an accuracy of 99%. This shows good handling capabilities with respect to the complexity of the UNSWNB datasets. This high accuracy, therefore, implies that the proposed model can make full use of the synergetic interaction between classification and optimization. This will improve detection capabilities beyond those of other methods.

In Figure 23, the performance of various hybrid classification and integrated optimization techniques on the UNSWNB dataset with respect to specificity is illustrated. It represents the true-negative rate and, therefore, illustrates how well a model can classify non-malicious network traffic. This high specificity is very significant in reducing the false-positive rate. This is a common problem associated with most network intrusion detection systems and responsible for generating spurious alerts and decreasing system efficiency. It can be shown that the proposed model will outperform others like NN, SVM, and DT with near-perfect specificity. The proposed model will, thus, be very effective at telling apart normal network activities from threats or anomaly traffic, reducing false alarms and increasing system reliability.

In Figure 24, the Rand index, which is a measure of the agreement of clustering by different classification methods with the real labels in the dataset, is presented. Rand index values range from 0 to 1; 1 is perfect agreement. This can be an appropriate metric to use in evaluating quality clustering as a hybrid technique, which puts optimization into classification. It can be seen from the figure that the proposed model has a Rand index very close to 1, indicating excellent performance in aligning its classification results very close to the real distribution of the UNSW-NB dataset. This high level of agreement indicates that the proposed model is quite effective at forming meaningful clusters mirroring the actual distribution of data. This ensures robust and reliable classification results. In Figure 25, the FPR and FNR calculations with different hybrid classification integrated optimization techniques on the UNSW-NB dataset are presented. The FPR shows the percentage of normal traffic misclassified as malicious, and the FNR is the percentage of malicious traffic misclassified as normal; both should have a low value for an effective intrusion detection system. The figure illustrates that traditional techniques like NN, SVM, and DT have large values of FPR and FNR, thus showing higher misclassification tendency. The proposed model, in contrast, reduces FPR and FNR considerably to 0.015 and 0.101, respectively. It also illustrates how the suggested model can provide better performance in terms of classifying network traffic in a way that will increase the reliability of the security system by reducing false alerts and missed detections. Optimization techniques are probably embedded within the proposed model that fine-tune classification boundaries, achieving superior detection accuracy.

The proposed work has many significant benefits, especially for security and adaptability in CPS. It integrates FedRCNet with ELFO, which allows high-accuracy threat detection through privacy-preserving decentralized learning to a certain extent. Adaptability to new and changing threats, along with the scalability feature, makes the model suitable for diverse and very different CPS environments, from industrial control systems to smart grids. There are also some limitations associated with this work. Such scenarios will impact model performance during local updates. This is based on the fact that federated learning will be ineffective when the data distribution across nodes is extremely non-uniform or some nodes are intermittently disconnected. Moreover, the enhanced efficiency in optimization may not help in highly complex or unknown attack patterns, which require more sophisticated model structures. Hence, despite its very promising approach, it requires further refinement and testing in more varied CPS scenarios to fully determine its robustness and scalability.

## 5. Conclusions

In this study, we propose the CPS ShieldNet Fusion model as an advanced architecture for improving the security settings of a cyber–physical system. With the increasing size, integration, and sophistication of cyber–physical systems, the focus of this study was to develop and validate a robust security framework using state-of-the-art techniques in federated learning and optimization that would effectively detect and mitigate the threat space in CPS environments. In these regards, the proposed CPS ShieldNet Fusion model combines Federated Residual Convolutional Networks and the EEL-Levy Fusion Optimization method for decentralized learning with accurate optimization to ensure the rightful detection of complex threats without exposing data privacy and security. The CPS ShieldNet Fusion model presents enormous potential for CPS cybersecurity by handling the challenges arising from these systems in a dynamic and distributed way. First, FedRCNet embedded in it enables an efficient training process across decentralized nodes. This ensures data privacy and reduces the risks related to their leakage. Second, by fine-tuning parameters, the model’s performance is refined by the ELFO method, which is very beneficial for improving its detection capability. Such a dual approach further strengthens CPS’s security posture and ensures the deliverance of a scalable solution that adapts to different network conditions and newly emerging threats.

We used three major datasets, CICIoT-2023, Edge-IIoTset2023, and UNSWNB, to test the CPS ShieldNet Fusion model’s performance efficiency. These datasets cover almost all attack scenarios and system configurations, providing the most complete testbed to check the model’s performance. These results demonstrate an overall better ability to detect and classify cyber threats across different CPS environments. This is done with very high accuracy, precision, recall, and F1-scores. The model achieves near-perfect ROC values of 0.99 and above, showing its superior ability at separating benign and malicious activities. These are general findings that illustrate the potential of CPS ShieldNet Fusion for boosting cyber–physical systems’ security by a manifold margin. By integrating federated learning with the latest optimization techniques, this model not only enhances the current security challenges of CPS but also acts as the basis for future research and development in this area. It contributes one essential step toward developing resilient, adaptive security solutions to keep pace with the fast-evolving threat landscape within CPS environments.

Moreover, the advanced optimization techniques built into the system would improve its detection capability for complex anomalies and make it adaptable to changing conditions. This will lead to a reduction in response times and a more effective security posture. This framework opens up opportunities for CPS security solutions, scalable and applicable to smart grids in healthcare and industrial control systems. These solutions have critical needs for data privacy and real-time threat detection.

## Figures and Tables

**Figure 1 sensors-25-03617-f001:**
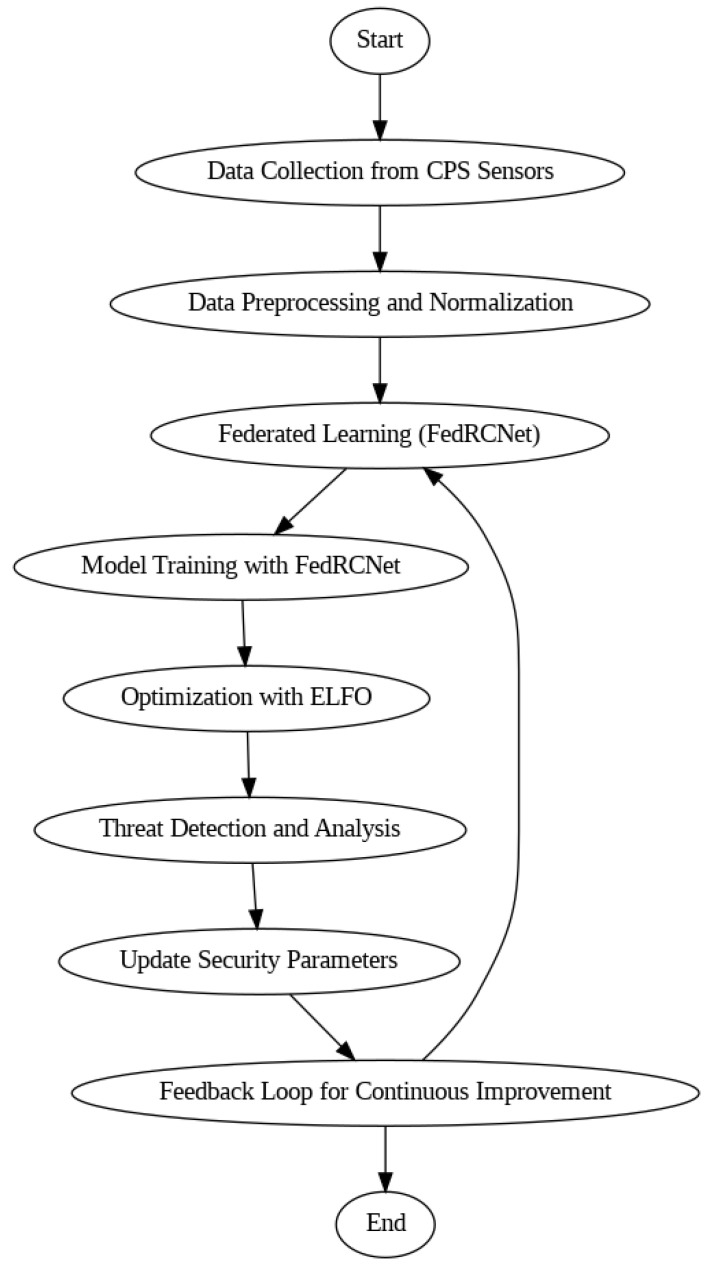
Overall flow of the proposed CPS ShieldNet Fusion model.

**Figure 2 sensors-25-03617-f002:**
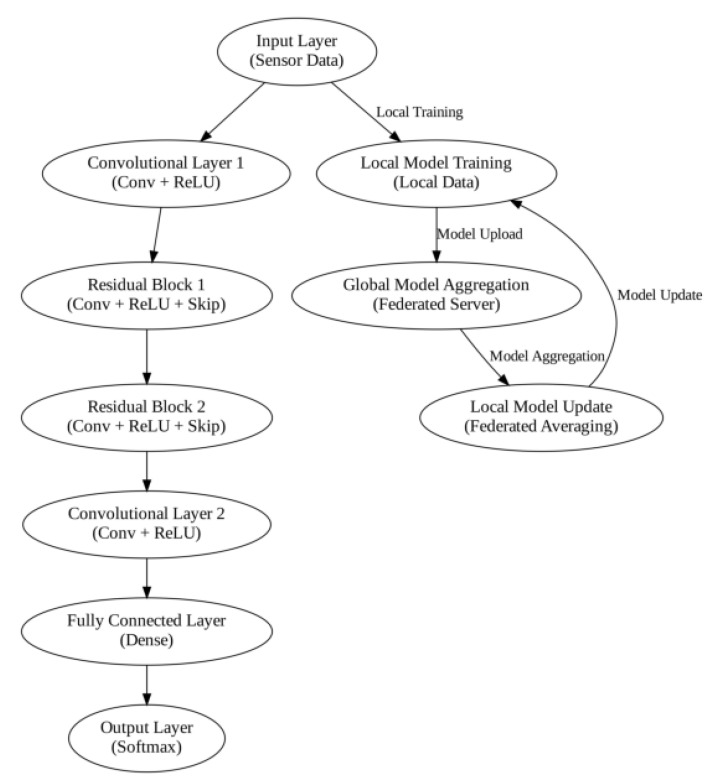
Architecture of proposed FedRCNet model.

**Figure 3 sensors-25-03617-f003:**
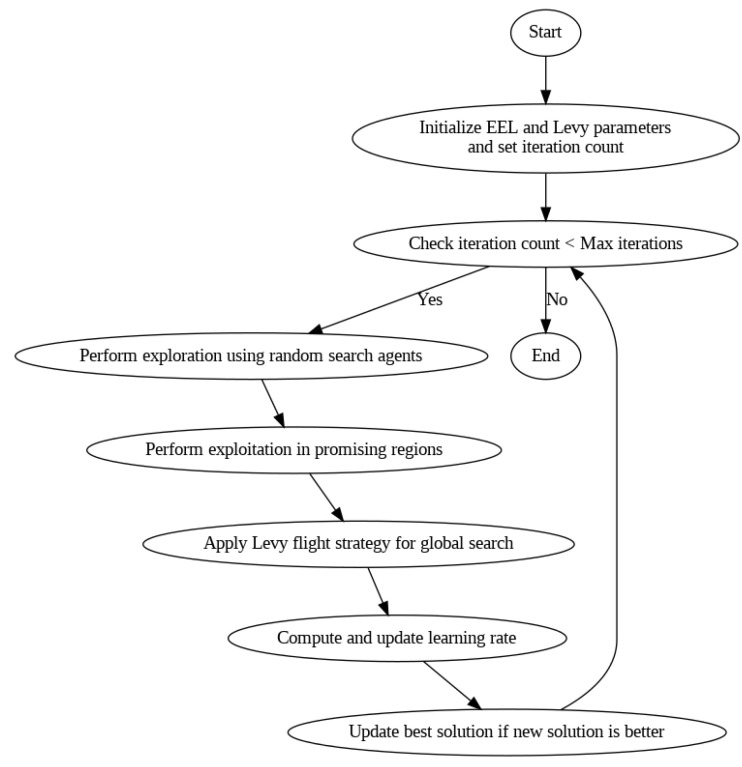
Flow of the proposed ELFO model.

**Figure 4 sensors-25-03617-f004:**
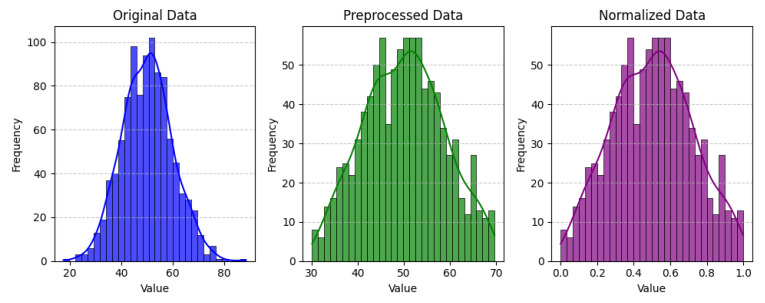
Histogram of original and normalized data.

**Figure 5 sensors-25-03617-f005:**
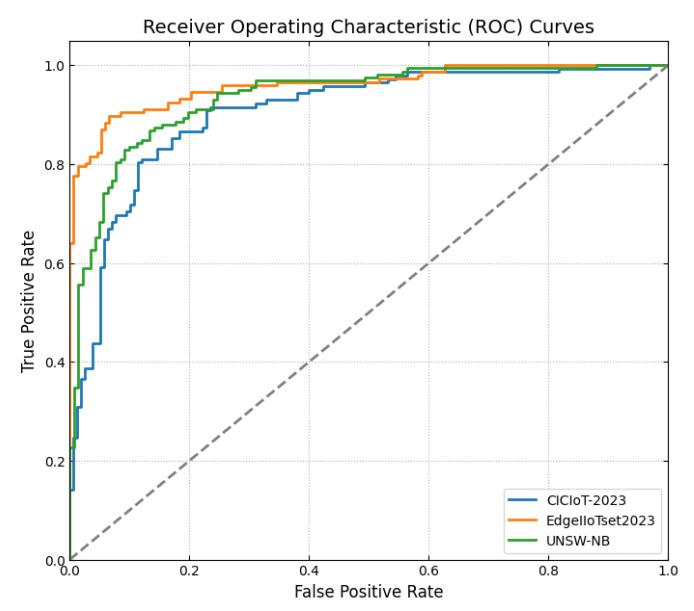
ROC for 3 datasets.

**Figure 6 sensors-25-03617-f006:**
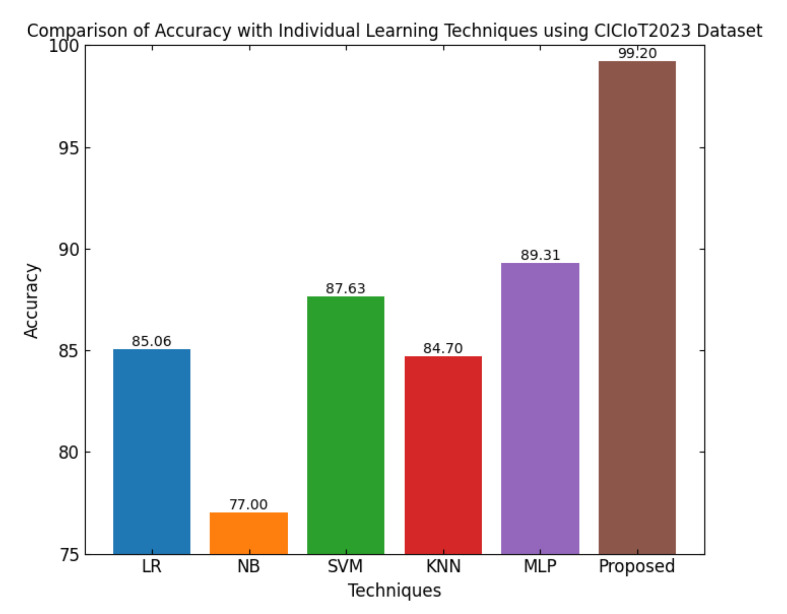
Comparison of accuracy with individual machine learning techniques using CICIoT2023 dataset.

**Figure 7 sensors-25-03617-f007:**
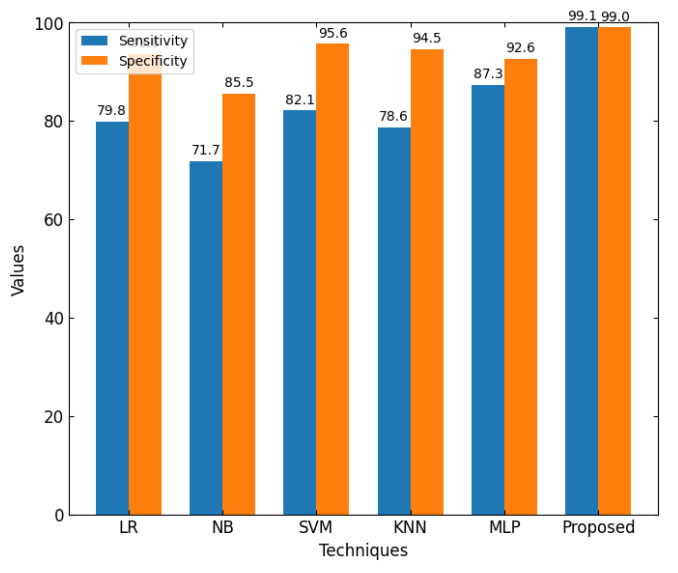
Comparison of sensitivity and specificity with individual machine learning techniques using CICIoT2023 dataset.

**Figure 8 sensors-25-03617-f008:**
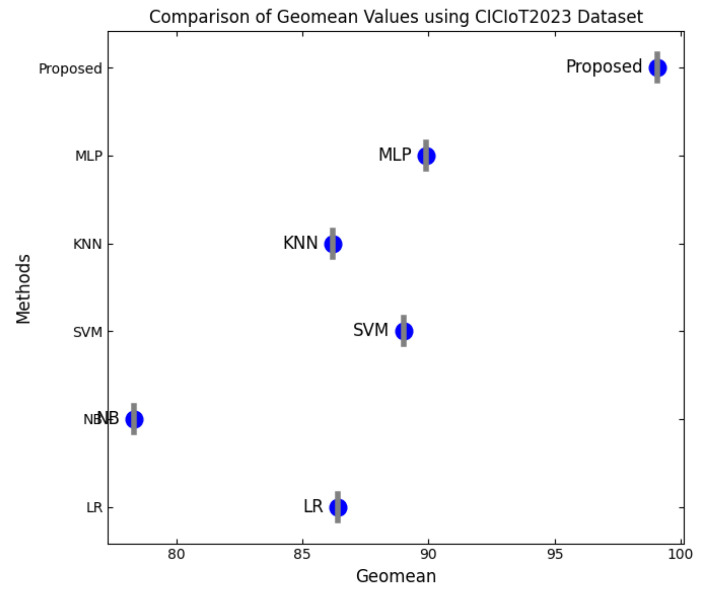
Comparison of G-mean with individual machine learning techniques using CICIoT2023 dataset.

**Figure 9 sensors-25-03617-f009:**
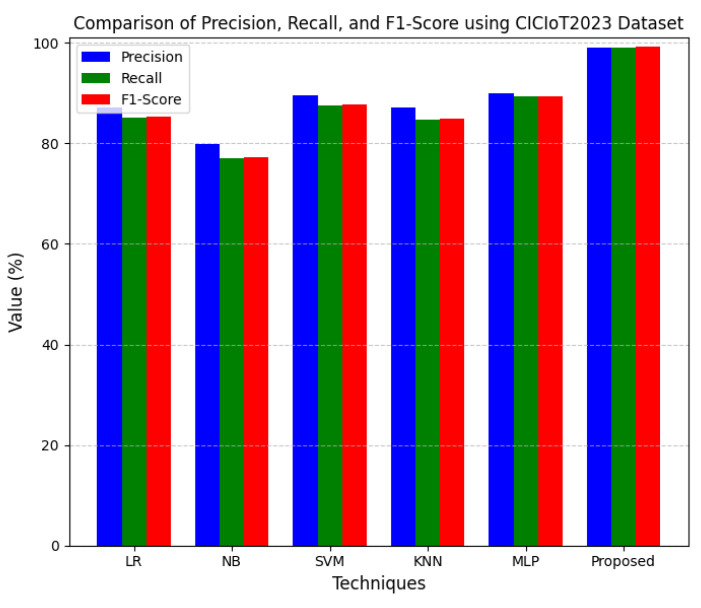
Comparison of precision, recall and F1-score with individual machine learning techniques using CICIoT2023 dataset.

**Figure 10 sensors-25-03617-f010:**
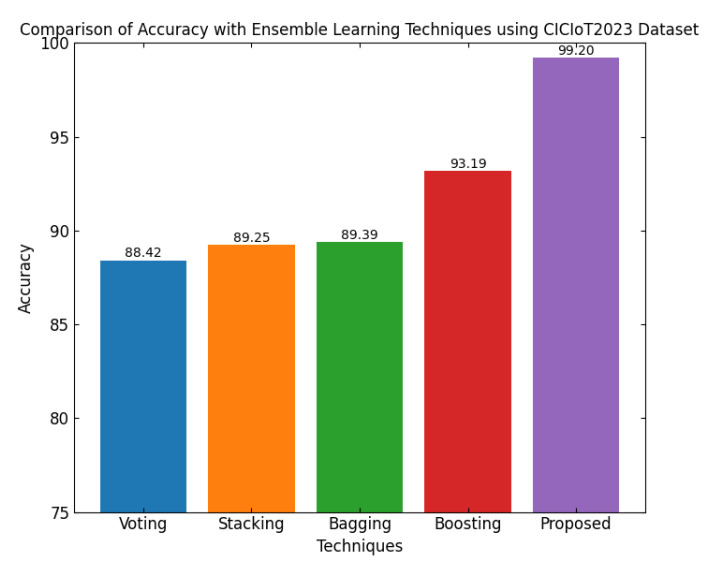
Comparison of accuracy with ensemble learning techniques using CICIoT2023 dataset.

**Figure 11 sensors-25-03617-f011:**
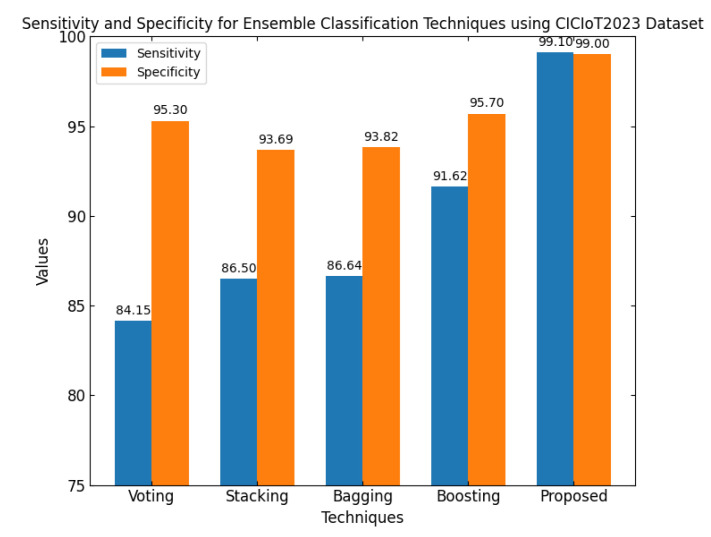
Comparison of sensitivity and specificity with ensemble learning techniques using CICIoT2023 dataset.

**Figure 12 sensors-25-03617-f012:**
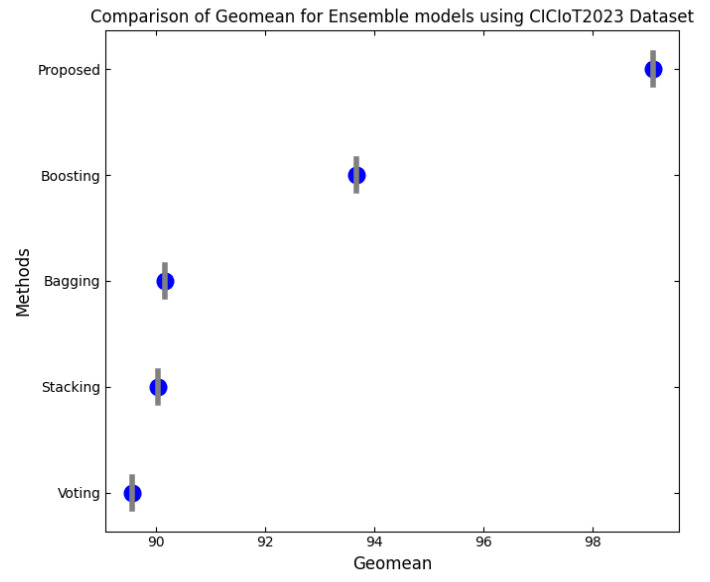
Comparison of G-mean with ensemble learning techniques using CICIoT2023 dataset.

**Figure 13 sensors-25-03617-f013:**
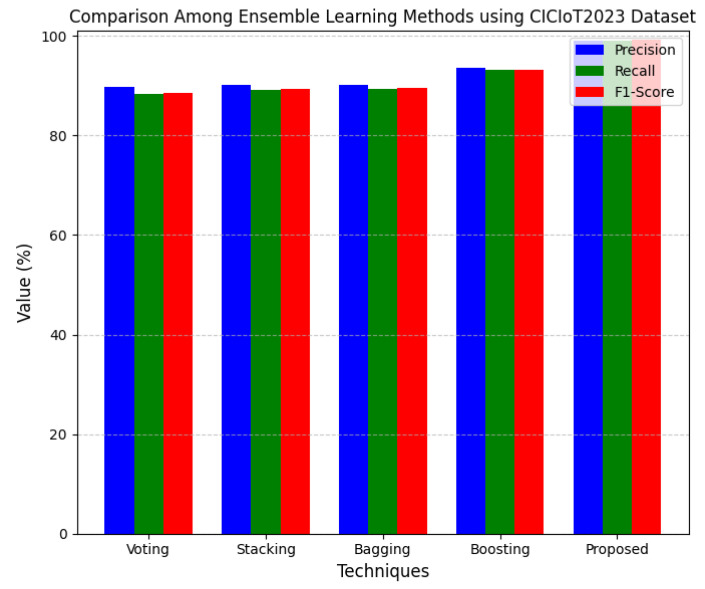
Comparison of precision, recall and F1-score with ensemble learning techniques using CICIoT2023 dataset.

**Figure 14 sensors-25-03617-f014:**
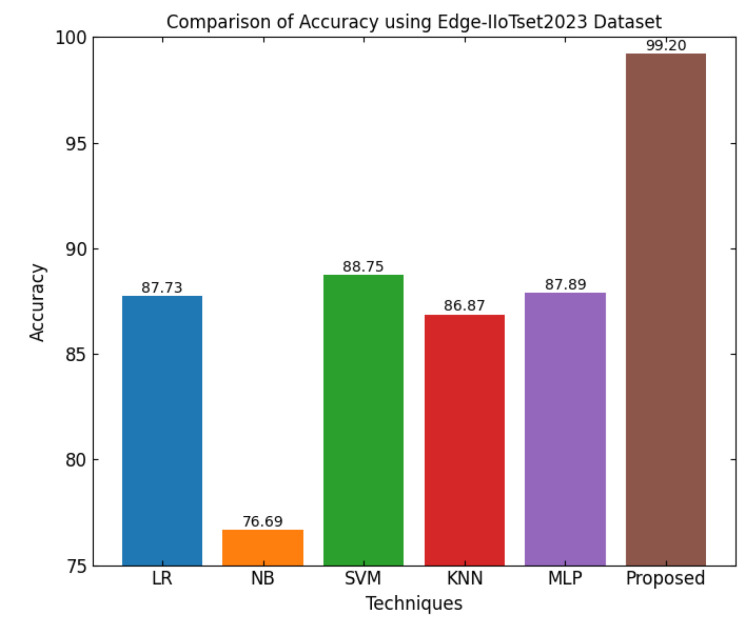
Comparison of accuracy with individual machine learning techniques using Edge-IIoTset2023 dataset.

**Figure 15 sensors-25-03617-f015:**
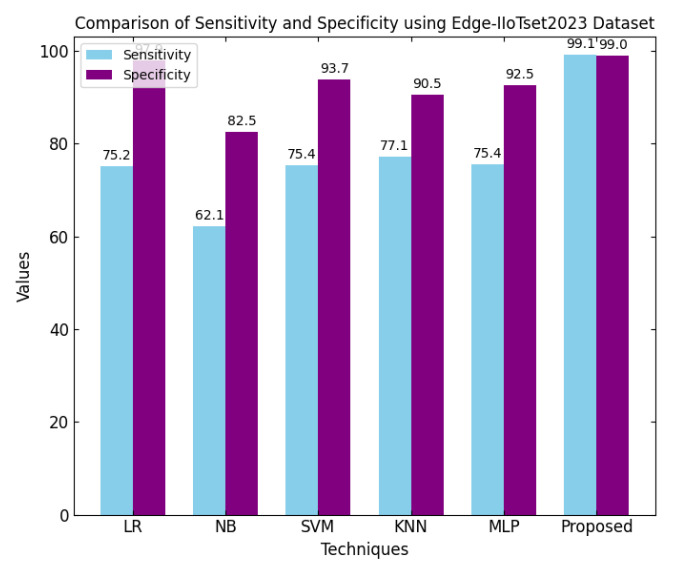
Comparison of sensitivity and specificity with individual machine learning techniques using Edge-IIoTset2023 dataset.

**Figure 16 sensors-25-03617-f016:**
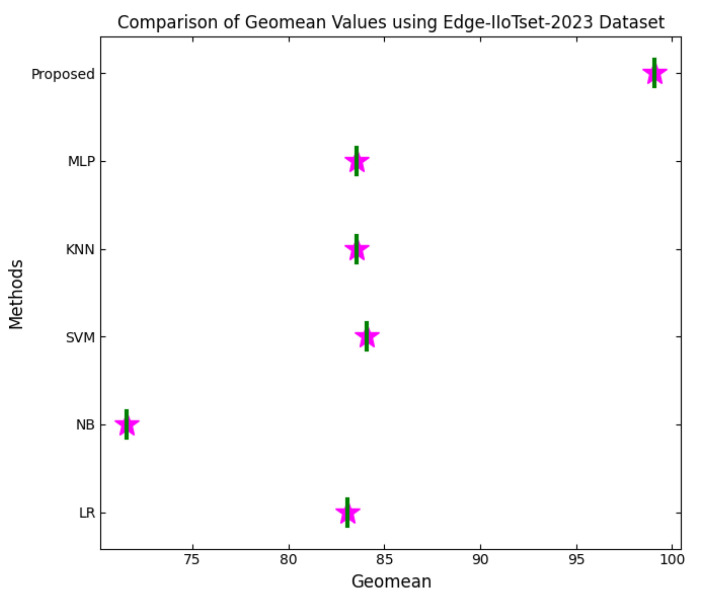
Comparison of G-mean with individual machine learning techniques using Edge-IIoTset2023 dataset.

**Figure 17 sensors-25-03617-f017:**
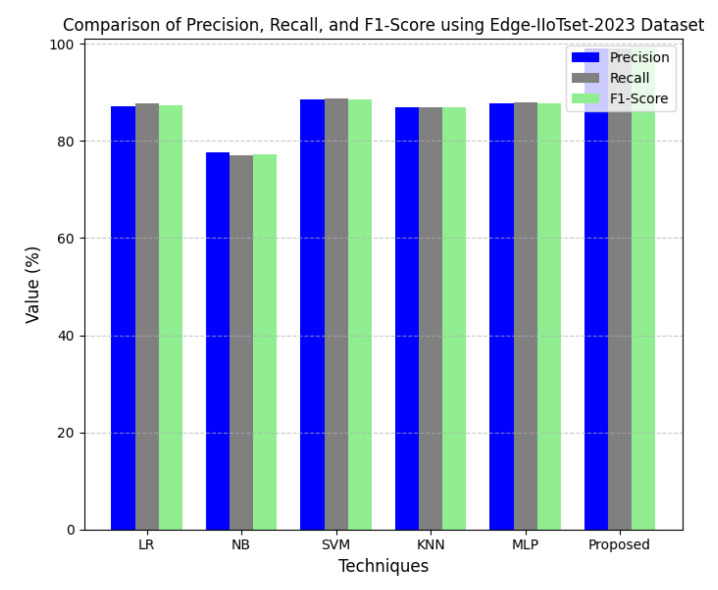
Comparison of precision, recall and F1-score with individual machine learning techniques using Edge-IIoTset2023 dataset.

**Figure 18 sensors-25-03617-f018:**
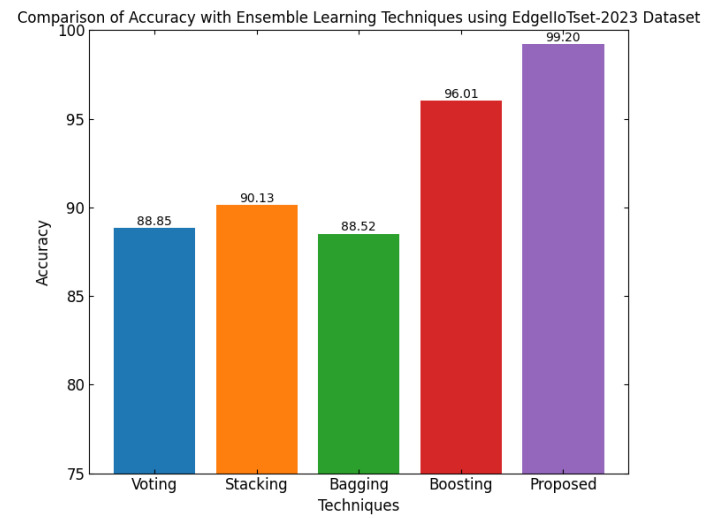
Comparison of accuracy with ensemble learning techniques using EdgeIIoTset2023 dataset.

**Figure 19 sensors-25-03617-f019:**
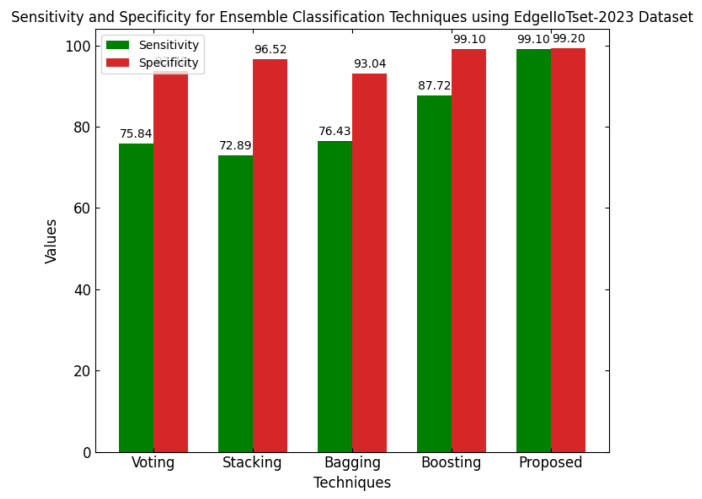
Comparison of sensitivity and specificity with ensemble learning techniques using EdgeIIoTset2023 dataset.

**Figure 20 sensors-25-03617-f020:**
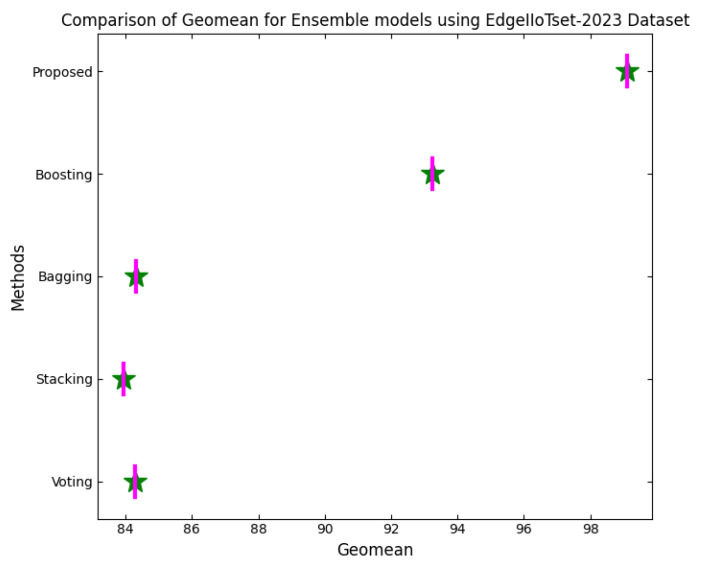
Comparison of G-mean with ensemble learning techniques using EdgeIIoTset2023 dataset.

**Figure 21 sensors-25-03617-f021:**
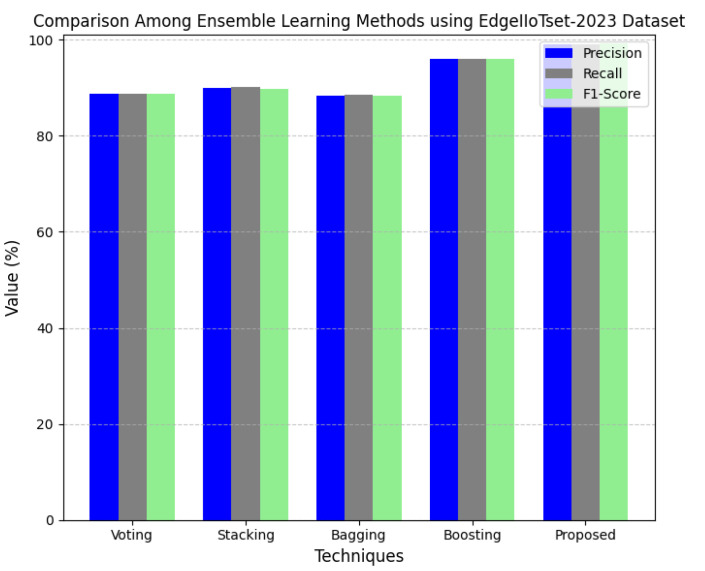
Comparison of precision, recall and F1-score values with ensemble learning techniques using EdgeIIoTset2023 dataset.

**Figure 22 sensors-25-03617-f022:**
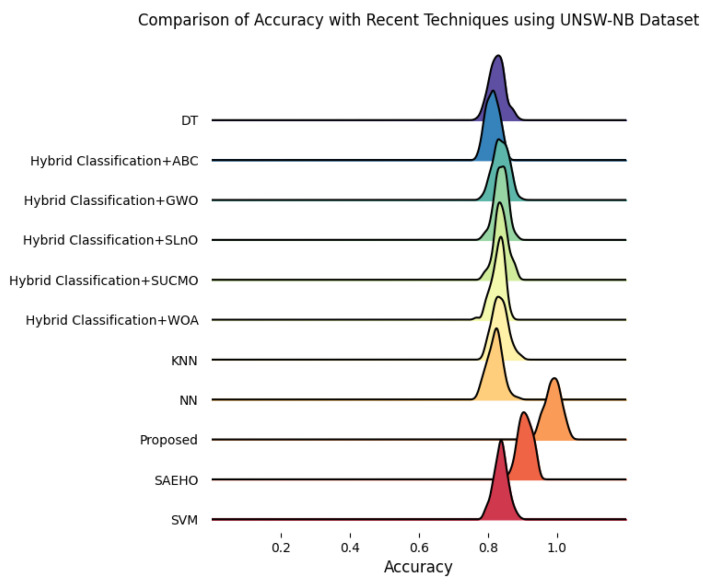
Comparison of accuracy with recent state of the art hybrid classification integrated optimization techniques using UNSW-NB dataset.

**Figure 23 sensors-25-03617-f023:**
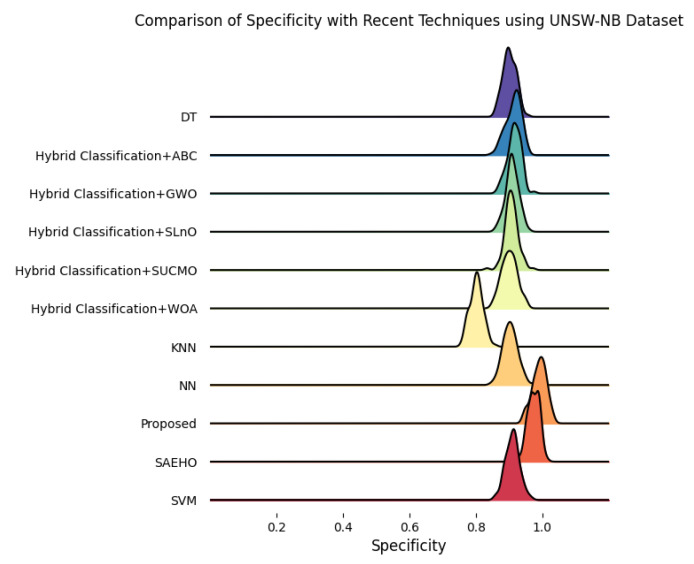
Comparison of specificity with recent state of the art hybrid classification integrated optimization techniques using UNSW-NB dataset.

**Figure 24 sensors-25-03617-f024:**
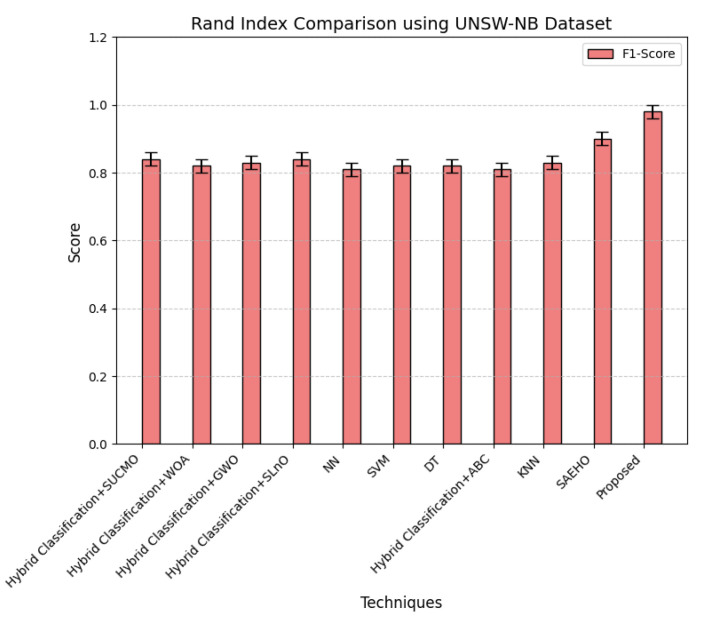
Comparison of Rand index with recent state-of-the-art hybrid classification integrated optimization techniques using UNSW-NB dataset.

**Figure 25 sensors-25-03617-f025:**
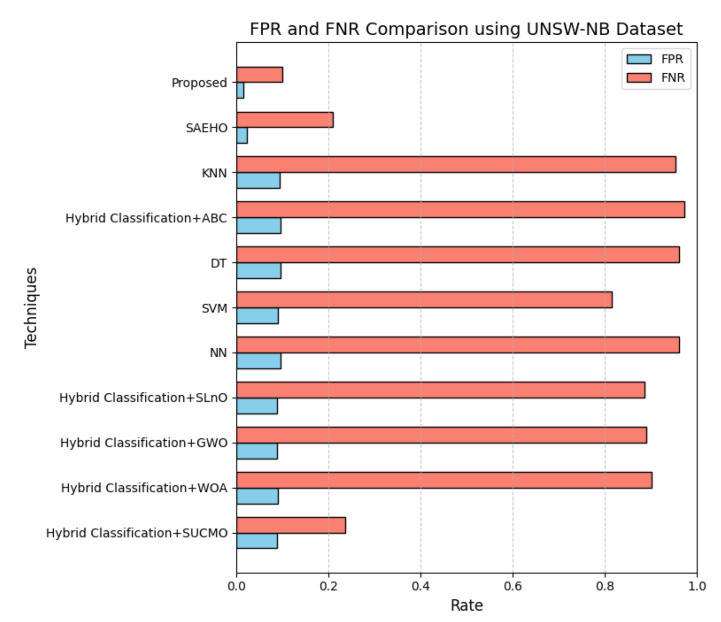
Comparison of FPR and FNR with recent state-of-the-art hybrid classification integrated optimization techniques using UNSW-NB dataset.

## Data Availability

The original data presented in the study are openly available in [www.kaggle.com] (accessed on 28 January 2025).

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
