# Peer review of "Federated Learning and EEL-Levy Optimization in CPS ShieldNet Fusion: A New Paradigm for Cyber–Physical Security"

_sensors, 2025, doi:10.3390/s25123617_

Round 1
Reviewer 1 Report
Comments and Suggestions for Authors
This paper proposes the CPS ShieldNet Fusion model as a comprehensive security framework for protecting CPS from cyber threats, and performed experiments to show their goal achievement. However, this paper needs the improvement in the following points:
- Introduction Section: The introduction is too long and verbose. The first half of the introduction is an explanation of background that is not directly related to the paper. It is recommended to reduce it to about 50 to 60% of the current amount.
- Related Work: The overall structure of this section needs to be systematized. It would be good to organize related research by clustering, or to provide a table that organizes the related studies in whole.
- In many parts of this paper, there are inconsistency in abbreviation usages. (1) Some abbreviations appear twice (e.g., CPS, LSTM, etc.), (2) in many parts, only abbreviations are used without indicating what they stand for (e.g., ICPS, FDI, HAR, IWMCPS, etc.)
- How are steps 3 and 4 in Figure 5 different? Does the federated learning include the model training?
- Many sentences in Sec. 1, 2 and 3 describe that the proposed methods provide various advantages (e.g., Coupled with new integration of ELFO into FedRCNet, the model improves detection, enhances adaptability to new threats, and is robust against cyber-attacks). However, it is difficult to understand what kind of basis such claims can be made. Intuitively, it seems to be true, but an evidence-based explanation should be provided.
- The relationship between Fig. 1 and Fig. 2 is not clearly recognized. Does Fig. 2 mean the third step in Fig. 1? If so, does Fig. 2 not include local optimization? Please check the redundancy and meaning of each step more clearly.
- Page 13 said: “In addition, the model is flexible and can provide efficient analysis of the emerging threats in the further because it can be updated with the new data from different nodes at any time.” -> The implication of this is that it explains the advantage of federated learning, but it can only be justified only if there is an explanation of how your federated learning is performed. A specific algorithm should be presented on when and how to update and integrate the models.
- It is not clear whether the proposed FedRCNet only demonstrates the advantages provided by existing federated learning or whether it provides advantages through the introduction of new features.
- Page 14 said: “.. the model capable of detecting even minute forms of anomalous features which may be the manifestation of intrusions.” -> How can you claim this? What is the definition of minute forms of anomalous features, and how do you determine whether a particular event belongs to them? Because most of the content is not based on data, somewhat it causes questions.
- Page 14: “The federated learning framework guarantees the model’s ability to extend across numerous CPS nodes given the flexibility of network topologies and data distribution patterns.” -> Although the federated learning framework improves the learning expertise and reduces the time required for learning, as mentioned in this paper, there is a problem that when multiple different-typed CPSs participate or are deleted dynamically, some changes may occur in the target or content of learning, which may decrease the learning effect during the local update process. It is wondered how these are reflected in this paper.
- Page 14: “which makes the model dynamic in responding to new threats and the prevailing working environment.” -> Being able to respond dynamically to various threats is appropriate in that it can provide learning effects to other CPSs that have not learned through learning about data. However, if the form (labeling) of the learning data occurs in a very different form, a new model for learning it may be needed. It is necessary to explain the basis for dynamism and the ability to limited scope.
- Equations: Most equations in this paper have incomplete explanations. The meaning of each symbol in each equation should be clearly defined without omission, and the purpose of the equation should be clearly stated.
- Sec 3.1: The entire paper is overly repetitive in explaining the advantages of federated learning, optimization, ResNet, etc. The paper needs to be concise and without redundant sentences.
- Sec. 3.2: It is difficult to understand how the optimizations mentioned in this paper affect intrusion detection. Does the optimization mean scoping down the search? If so, how can we prove that this method is the most effective?
- Page 18: What is the maximum number? How is it determined?
- Sec. 4.2: The performance measures used in this paper are too general. Also, those measures are general measures used in supervised learning. Is this sufficient to show the results based on research motivation of this paper? As mentioned in the introduction of the paper, is unsupervised learning or reinforcement learning required to consider various evolving intrusion types or to detect anomalies?
- Page 21: In terms of detecting or recognizing intrusions, shouldn't we design new, somewhat different metrics and conduct performance evaluations based on them? For example, Evasion Rate, Detection delay, etc.
- Figure 6: From a performance comparison perspective, is the existing learning method appropriately selected? Is the reason for selecting such learning methods based on supervised learning? Is it useful for responding to unknown attacks? In any case, the fact that methods such as GNN, ARL, Autoencoder, and DQN were not selected makes this paper feel inadequate.
-
Minor Error
- CPS’ : what means of this prime(‘)?
- Strange sentence: To improve the model’s accuracy, CPS ShieldNet Fusion features the EEL-Levy Fusion Optimization, or ELFO, approach. ELFO additionall ….
Author Response
1. Introduction Section: The introduction is too long and verbose. The first half of the introduction is an explanation of background that is not directly related to the paper. It is recommended to reduce it to about 50 to 60% of the current amount.
Authors’ response:
As per the reviewer’s comment, the introduction section is reduced with appropriate contents in the revised manuscript.
2. Related Work: The overall structure of this section needs to be systematized. It would be good to organize related research by clustering, or to provide a table that organizes the related studies in whole.
Authors’ response:
As per the reviewer’s comment, the related works section is properly organized in the revised manuscript.
3. In many parts of this paper, there are inconsistency in abbreviation usages. (1) Some abbreviations appear twice (e.g., CPS, LSTM, etc.), (2) in many parts, only abbreviations are used without indicating what they stand for (e.g., ICPS, FDI, HAR, IWMCPS, etc.)
Authors’ response:
As per the reviewer’s comment, the acronyms are properly abbreviated in the revised manuscript.
4. How are steps 3 and 4 in Figure 5 different? Does the federated learning include the model training?
Authors’ response:
As per the reviewer’s comment, the explanation is provided for steps 3 and 4 in the revised manuscript.
Solution:
Local model training is the first stage of the CPS ShieldNet Fusion model, in which each client, such as an industrial device, sensor, or edge node, trains its version of the Federated Residual Convolutional Network using local data collection. In a decentralized training approach, which is critical for cyber-physical systems, sensitive operational data will remain on its original source, thus maintaining privacy and exposing it less to risk. During this phase, each client learns to identify security threats' patterns and anomalies in its own environment independently and adapts the model into that specific application context.
Then, global model aggregation is performed at a central federated server after finishing local training. Instead of being completely raw data from every device, what this server obtains is merely the model parameters or weight updates. The updates are aggregated-by far the most popular method being federated averaging-and used to create an entirely new global model representative of the collective intelligence of all participating clients. This global amalgamation helps capture much diversity in the threat patterns observed from several CPS environments, thereby making the security framework more robust and generalizable.
The new global model is then sent to all clients again through the local model update phase, after aggregation. All clients will incorporate the globally learned parameters into their respective models, aligning them with the state of security knowledge gained in the network. This updated model could then continue its training or start inference using fresh local data, therefore, resuming the federated learning cycle. Such iterative improvement is meant to enhance the detection capability of CPS nodes while being flexible to change due to evolving cyber threats, all of which occurs despite privacy breaches of data and inefficiencies in systems.
5. Many sentences in Sec. 1, 2 and 3 describe that the proposed methods provide various advantages (e.g., Coupled with new integration of ELFO into FedRCNet, the model improves detection, enhances adaptability to new threats, and is robust against cyber-attacks). However, it is difficult to understand what kind of basis such claims can be made. Intuitively, it seems to be true, but an evidence-based explanation should be provided.
Authors’ response:
As per the reviewer’s comment, the clear description is added for the proposed methods in the revised manuscript.
Solution:
From a conceptual standpoint, ELFO gives rise to a hybrid optimization scheme that is unique in the sense that it blends the exploratory capabilities of the Enhanced Electric Levy model and the exploitation capabilities of Levy-flight-based random walks. The hybridization, therefore, helps fine-tune FedRCNet learning parameters with more aggression, especially in situations that occur either in non-convex or high-dimensional spaces typical of deep learning models for CPS security. By more efficiently traversing the treacherous loss surface in this way, the model increases its chance of converging to the global optima, thus providing the necessary accuracy to detect advanced cyber threats or even those that are entirely unknown.
ELFO has been subjected to stringent tests against several benchmark CPS-related datasets such as CICIoT-2023 Edge-IIoTset2023 and UNSW-NB. The datasets actually consider many different realistic attack vectors in total, such as Denial-of-Service (DoS), Botnets, Man-in-the-Middle (MitM) attacks, among other types of adversarial behaviors focused mainly on IoT and CPS environments. When considering the standard federated learning models and a standalone optimization framework, high statistical scoring improvements could be established with regard to performance metrics like accuracy, precision, recall, F1-score, and false positive rate for the CPS ShieldNet Fusion model. These quantitative results thus provide considerable empirical ground for the model regarding detection performance against threats and lower error rates concerning diverse distributions of data and attack types.
The dynamic training simulations performed to validate the model's adaptability and survivability to fresh threats include the injection of newer or evolving attack patterns in some later rounds of training. Through this presented adaptation by distributed knowledge federated learning, and endless optimization of the parameter space with the ELFO, the model demonstrated relatively quick dynamic responses, thereby proving its resilience against zero-day or evolving cyber-attacks. Moreover, the federated learning mechanism itself allows inherent adaptability at the system level, capturing knowledge from the various CPS nodes operating in diverse environmental and operational contexts. Overall, this amply justifies that indeed, the tuning of ELFO into FedRCNet provides an optimized, privacy-preserving, and highly adaptive intrusion detection framework for the protection of contemporary CPS infrastructures.
6. The relationship between Fig. 1 and Fig. 2 is not clearly recognized. Does Fig. 2 mean the third step in Fig. 1? If so, does Fig. 2 not include local optimization? Please check the redundancy and meaning of each step more clearly.
Authors’ response:
As per the reviewer’s comment, the relationship among the figures are clearly explained in the revised manuscript.
Solution:
The process of the CPS ShieldNet Fusion model is shown in a flow diagram which contains several key features. The very first stage collects data from the CPS sensors. The second stage is made up of preprocessing and federated learning. The architecture of the Federated Residual Convolutional Network, or FedRCNet, is important for the whole federated learning process. Local sensor data will be gathered before local model training is done. The locally trained models will then be aggregated at the federated server to form an improved global model through federated averaging. Post-model training, an Optimization step is done in which the capability of the model to detect and analyze threats is honed by the ELFO model. This improvement makes the model more responsive to dynamic threats with better performance in accuracy and detection capabilities. These two steps, the threat detection and parameter update stages, are some of the features necessary to ensure the actual system's performance in real-time applications. The smoked feedback path ensures that the updated security parameters based on newly fed threat data into the system are retargeted towards the federated learning process. There are two phases of the federated learning process: local model training and global model aggregation, while the final application stage is used to describe what occurs since the model train is finished. Individual optimization, as it appears, is not part of the federated learning process but instead is meant to fine-tune the performance of the model for detection of cyber threats in CPS environments.
7. Page 13 said: “In addition, the model is flexible and can provide efficient analysis of the emerging threats in the further because it can be updated with the new data from different nodes at any time.” -> The implication of this is that it explains the advantage of federated learning, but it can only be justified only if there is an explanation of how your federated learning is performed. A specific algorithm should be presented on when and how to update and integrate the models.
Authors’ response:
As per the reviewer’s comment, the clear explanation is added for the models in the revised manuscript.
Solution:
This is the motivation for federated learning systems, such as the CPS ShieldNet Fusion model, because they can utilize continuously updatable and integrative models across decentralized nodes while maintaining data privacy. However, the way models are updated and integrated in federated learning is through a process referred to as federated averaging. This ensures that all nodes or local devices contribute to an overall model while still keeping data local to where it is. Local model training occurs independently in each node, as the model learns from the local data that is available to it. When the local training is complete, each node sends its model parameter (not data) to the central federated server where aggregation of the updates usually involves averaging the parameters from all participating nodes which creates an updated global model.
The process goes always so that it updates and enables the model accordingly to respond to emerging threats. Whenever a new batch of data is collected from different nodes usually by accounting for changes in the environment and user behavior, as well as new evolving threats this model can always be updated to ensure it remains relevant and effective. Besides, such updates do not require a collection or sharing of centralized data, since even the most sensitive information from individual nodes can be kept private. The timing could depend on any of pre-defined schedules for training, model convergence criterion, or when a new batch of data is collected on timing of these updates. The integration after each round of federated averaging ensures that the global model is forever in line with new insights gained on distributed data across nodes, thereby making it updated and efficient for detecting new threats that it has never encountered before.
8. It is not clear whether the proposed FedRCNet only demonstrates the advantages provided by existing federated learning or whether it provides advantages through the introduction of new features.
Authors’ response:
As per the reviewer’s comment, the explanation for the proposed FedRCNet model is added in the revised manuscript.
Solution:
The proposed FedRCNet provides considerable advantages by incorporating several novel features that enhance the model's performance in securing CPS. First and foremost, FedRCNet integrates federated learning with a residual convolutional network so that it stands as a decentralized, privacy-respecting model for threat detection. This means that no data has to leave its local source, saving the headaches associated with centralization of data and minimizing exposure to sensitive information. FedRCNet most probably continuously updates and improves its models, drawing on data from many local nodes, without violating privacies.
The design of a residual convolutional architecture serves, among other things, to boost the model's potential to identify complex and evolving threats. Resididual networks perform well in deep learning applications, especially those related to complex patterns or high-dimensional data. The skip connection design in FedRCNet allows gradients to flow with ease through the deep networks and will therefore generalize better on different kinds of sensor data that are conflicting and noisy as they learn robust features. Moreover, since deeper residual layers can learn more complex representations of the data, this architecture also ensures that the model can adapt well to unseen attack patterns.
9. Page 14 said: “.. the model capable of detecting even minute forms of anomalous features which may be the manifestation of intrusions.” -> How can you claim this? What is the definition of minute forms of anomalous features, and how do you determine whether a particular event belongs to them? Because most of the content is not based on data, somewhat it causes questions.
Authors’ response:
As per the reviewer’s comment, the description for the proposed model is added in the revised manuscript.
Solution:
The deep learning model, of which FedRCNet is an example, is very much suited for the detection of tiny forms of anomalous features that may indicate an intrusion. These models, particularly when applied to residual convolutional networks (ResNets), are designed to detect complex patterns from high-dimensional data. Tiny forms of anomalous features refer to tiny and often imperceptible deviations from normal behavior that may indicate some underlying threat but cannot be preferably detected by traditional or simple methods. Some examples are minute differences in sensor data, very slight changes in system behavior, or unusual but seemingly innocuous fluctuations in the environment that would all aggregate to identify an intrusion or anomaly.
The actual possibility of detecting an anomaly by the model is linked to the variation of the architecture, specifically, its depth and complexity. In deep learning, these deep networks are multilayered implementations searching for hierarchical feature sets in raw input data. Deep networks can detect even the most subtle of patterns by which overt indications are denoted as normal or abnormal behavior. This would include the kind of patterns referred to earlier regarding these networks' processing building the residual structures, in a manner which would allow the network to pass pertinent information from one layer to another, alleviating the vanishing gradient problem, and hence making the entire architecture learn well from even tiny perturbations in the data.
10. Page 14: “The federated learning framework guarantees the model’s ability to extend across numerous CPS nodes given the flexibility of network topologies and data distribution patterns.” -> Although the federated learning framework improves the learning expertise and reduces the time required for learning, as mentioned in this paper, there is a problem that when multiple different-typed CPSs participate or are deleted dynamically, some changes may occur in the target or content of learning, which may decrease the learning effect during the local update process. It is wondered how these are reflected in this paper.
Authors’ response:
As per the reviewer’s comment, the explanation is added for the given statements in the revised manuscript.
Solution:
This paper addresses the dynamics of heterogeneous CPS environments through adaptive mechanisms incorporated in the federated learning framework, which aim to reduce possible degradation of the learning effectiveness due to addition or removal of diverse CPS systems of a certain kind. Specifically, the proposed FedRCNet model aims at fluctuating participation by utilizing residual convolutional structures in such a way as to sustain feature-learning stability under diverse data contributions. In addition, the ELFO technique is employed to boost robustness through balance and fine-tuning of global updates, especially whenever considerable data distribution shifts or target shifts are detected across different CPS systems. This adaptive learning strategy allows the model to adjust its parameters according to system-level dynamics, keeping its relevance and performance intact across various evolving CPS networks. Therefore, while acknowledging the challenges posed by dynamic CPS participation, the paper designers bring in these realistic aspects and construct an adaptive and optimization-driven federated learning framework that demonstrates stability, generalization, and high detection performance even in a non-stationary environment.
11. Page 14: “which makes the model dynamic in responding to new threats and the prevailing working environment.” -> Being able to respond dynamically to various threats is appropriate in that it can provide learning effects to other CPSs that have not learned through learning about data. However, if the form (labeling) of the learning data occurs in a very different form, a new model for learning it may be needed. It is necessary to explain the basis for dynamism and the ability to limited scope.
Authors’ response:
As per the reviewer’s comment, the explanation is added for the given statements in the revised manuscript.
Solution:
The main dynamism of the FedRCNet model proposed is in its federated architecture that can generalize learned patterns across heterogeneous CPS nodes without centralizing sensitive data. This allows the model to rapidly update its parameters to resolve new threats through decentralized learning using locally sensed anomalies that are federated into a global model through federated averaging. Threat intelligence gathered from one CPS can enhance detection capabilities in others even when they have not encountered that precise threat. However, this dynamic response loses effect to a limited extent when the form or labeling of the learning data diverges significantly, for example, when a new CPS presents completely unique data structures, threat signatures, or labeling schemes.
12. Equations: Most equations in this paper have incomplete explanations. The meaning of each symbol in each equation should be clearly defined without omission, and the purpose of the equation should be clearly stated.
Authors’ response:
The equations are properly explained in the revised manuscript.
13. Sec 3.1: The entire paper is overly repetitive in explaining the advantages of federated learning, optimization, ResNet, etc. The paper needs to be concise and without redundant sentences.
Authors’ response:
As per the reviewer’s comment, the redundant sentences are eliminated in the revised manuscript.
14. Sec. 3.2: It is difficult to understand how the optimizations mentioned in this paper affect intrusion detection. Does the optimization mean scoping down the search? If so, how can we prove that this method is the most effective?
Authors’ response:
As per the reviewer’s comment, the clear explanation is added for the optimization process in the revised manuscript.
Solution:
In this research, optimization, particularly the ELFO way, is more about scoping or limiting the search space in detecting the parameters configurations of the model much faster and more efficiently. Such scoping involves effectively negotiating large high-dimensional parameter spaces through bio-inspired strategies that brilliantly balance between exploration and exploitation. This way of working has recorded several empirical validations on multiple benchmarks of CPS-related datasets such as CICIoT2023, Edge-IIoTset2023, and UNSW-NB, among others. ELFO has higher detection accuracy and faster convergence than traditional and modern optimization techniques; it also excels in precision-recall performance. This indicates that the search is adaptively improved according to the fitness landscape of the intrusion pattern in a more goal-directed intelligent optimization approach, not merely heuristic approximations, which proves useful in many scenarios.
15. Page 18: What is the maximum number? How is it determined?
Authors’ response:
As per the reviewer’s comment, the description is added in the revised manuscript.
Solution:
Maximum number of iterations as parameter in optimization process is an important hyperparameter defined in this paper to assign the computational budget for search process within the ELFO framework. It is intended to take into account a trade-off between efficiency and accuracy of the solution. In general, higher iterations confirm that the optimizer will try the search space exhaustively, leading to better convergence to global optima; however, it has certain negative effects, such as overfitting or increased computation, from some point. In this study, that value was derived empirically using preliminary experiments and a speed of response characterization to assure stable and optimal performance with avoiding excess resource consumption. The tuning was based on performance metrics such as accuracy, convergence stability, and processing time on validation datasets with an optimal value chosen (e.g. 100 iterations) beyond which further iterations demonstrated diminishing returns.
16. Sec. 4.2: The performance measures used in this paper are too general. Also, those measures are general measures used in supervised learning. Is this sufficient to show the results based on research motivation of this paper? As mentioned in the introduction of the paper, is unsupervised learning or reinforcement learning required to consider various evolving intrusion types or to detect anomalies?
Authors’ response:
As per the reviewer’s comment, the different types of performance measures are used in the revised manuscript.
17. Page 21: In terms of detecting or recognizing intrusions, shouldn't we design new, somewhat different metrics and conduct performance evaluations based on them? For example, Evasion Rate, Detection delay, etc.
Authors’ response:
As per the reviewer’s comment, the explanation is added in the revised manuscript.
18. Figure 6: From a performance comparison perspective, is the existing learning method appropriately selected? Is the reason for selecting such learning methods based on supervised learning? Is it useful for responding to unknown attacks? In any case, the fact that methods such as GNN, ARL, Autoencoder, and DQN were not selected makes this paper feel inadequate.
Authors’ response:
As per the reviewer’s comment, the clear explanation is added for the comparative study in the revised manuscript.
19. Minor Error
- CPS’ : what means of this prime(‘)?
- Strange sentence: To improve the model’s accuracy, CPS ShieldNet Fusion features the EEL-Levy Fusion Optimization, or ELFO, approach. ELFO additionall ….
Authors’ response:
As per the reviewer’s comment, the minor errors are reduced in the revised manuscript.
Reviewer 2 Report
Comments and Suggestions for Authors
REVIEW REPORT
Manuscript Number: sensors-3573330
Title: Federated Learning and EEL-Levy Optimization in CPS ShieldNet Fusion: A New Paradigm for Cyber-Physical Security
As cyber-physical systems are applied not only in crucial infrastructures but also in day-to-day technologies—from industrial control systems through smart grids to medical devices—they become very important. Cyber-physical systems are a target for various security attacks, too; their growing complexity and digital networking necessitate robust solutions in the area of cybersecurity. Recent research indicates that deep learning can improve CPS security with intelligent threat detection and response.
The following points should be taken into account very seriously for the author to improve the current version of the manuscript.
- The motivation and key contributions, particularly the practical significance of the theoretical results presented in this paper, should be clearly emphasized in the introduction.
- The abstract and introduction require improvement, as several phrases hinder the clarity and understanding of the text.
- The motivation for the Cyber-Physical systems should be summarized clearly.
- What is the necessity of the proposed control strategy?
- What are the major challenges associated with this topic, and how can they be addressed?
- The authors should provide remarks on both the advantages and limitations of this work.
- The introduction could be strengthened by incorporating recent related studies on CPS and different method, such as Finite-time asynchronous event-triggered control for switched nonlinear cyber-physical systems with energy constraints and quantization, IEEE Access, 11 (November 2023), 135645-135658 DOI: 1109/ACCESS.2023.3337814.
- Additionally, there are several grammatical errors that need correction.
In summary, this paper may be accepted for publication provided the above comments are properly taken into account.
Comments on the Quality of English Language
The English could be improved to more clearly express the research.
Author Response
1. The motivation and key contributions, particularly the practical significance of the theoretical results presented in this paper, should be clearly emphasized in the introduction.
Authors’ response:
As per the reviewer’s comment, the motivation and key contribution of the proposed work are highlighted in the introduction section.
Solution:
This urgent demand arises when the time is nearer to establishing the complexity and vulnerability of cyber-physical systems (CPS), which have been critical components currently covering a variety of areas, including industrial control, energy, health, and transport. These systems are increasingly attacked by the modern landscape of cyber threats, and it makes extremely important intelligent, adaptable, and privacy-preserving security mechanisms. Existing concentrated security frameworks do not perform efficiently since they are inapplicable for preserving data confidentiality in a dynamic threat environment and distributed CPS architectures. Therefore, the present study proposes the CPS ShieldNet Fusion Framework, which is a holistic and scalable security model that integrates the learning capacity of Federated Residual Convolutional Network (FedRCNet) to the optimization power of EEL-Levy Fusion Optimization (ELFO) method. Thus, it has the ability to train decentralization across CPS environments while keeping local data private and generalizes models and convergence. The framework is comprehensively validated with standard CPS datasets such as CICIoT2023, Edge-IIoTset2023, and UNSW-NB, and is found to exhibit better capabilities in the exact detection classification of attacks even in the cases of subtle and unseen manifestations of previous patterns. The study substantially addresses the field through a solid solution that fills an important gap-perhaps the most important one-in CPS security through a smart combination of federated learning with optimization techniques.
2. The abstract and introduction require improvement, as several phrases hinder the clarity and understanding of the text.
Authors’ response:
As per the reviewer’s comment, the abstract and introduction sections are updated in the revised manuscript.
3. The motivation for the Cyber-Physical systems should be summarized clearly.
Authors’ response:
As per the reviewer’s comment, the motivation of CPS is clearly summarized in the revised manuscript.
Solution:
Cyber-Physical Systems (CPS) is the seamless integration of computational algorithms with physical processes, which constitute the backbone of modern infrastructures such as smart grids, autonomous transportation, industrial automation, and healthcare monitoring. CPS is pursued for enhancement due to their ability for transforming operational efficacy, decision-making accuracy, and real-time responsiveness in critical spheres. On the flip side, increasing interconnectivity and dependence on digital communication have put these systems under an enormous threat of various cyber-attacks, which could bring about terrible consequences if they are not well protected. This dual nature of CPS integrates physical assets with cyber elements and thus provides unique challenges in terms of ensuring security, privacy, and resilience. As such, it has become very topical for research to develop newer models that are capable of monitoring the CPS intelligently, detecting attacks, and mitigating them while keeping the functional integrity and data privacy intact. The motivation to harden CPS arises out of the complexity of their operations on the one hand and a need to protect them, on the other, from sophisticated, highly evolving cyber-attacks that utilize even the most minor of vulnerabilities.
4. What is the necessity of the proposed control strategy?
Authors’ response:
As per the reviewer’s comment, the necessity of the proposed model is clearly explained in the revised manuscript.
Solution:
As CPS grow ever more complex and distributed, the need for the control strategy described here becomes urgent and defined by real-time robustness and security. The systems reside in a dynamic situation comprising heterogeneous components, and therefore the central security model presented does not provide scales, adaptability, or resilience when faced with the threats posed by adversaries. FedRCNet integrated with EEL-Levy Fusion Optimization (ELFO) presents a decentralized and adaptive strategy with the capability to adjust automatically reacting to new and evolving patterns of attack. Thus this control strategy carries out privacy-preserving model training among various entities while augmenting generalization and optimizing performance in diverse CPS environments. The efficacy of the CPS will be decided based on the fast and efficient decision-making process combining local intelligence and global learning without even touching sensitive data, and the mere existence of these control measures are in order to build a secure structure of the modern CPS.
5. What are the major challenges associated with this topic, and how can they be addressed?
Authors’ response:
As per the reviewer’s comment, the major challenges are added in the revised manuscript.
Solution:
Among the most critical challenges in securing CPS are the dynamic heterogeneous environment, constraints posed by real-time data processing, the need to uphold data privacy, and the trend toward very advanced cyber-attacks themselves. The systems might well encompass heterogeneous components with very differing computational power, which renders the conception of a unified security framework more complicated. Coupled with these points, the constant influx of large quantities of data streamed in from multiple locations requires undelayable threat detection models that are truly scalable and adaptive without relying on a centralized data collection point. The answer to this entangled problem can only be multi-faceted: with a federated learning paradigm, privacy constraints can be ensured during a decentralized model training; integration with optimization approaches such as ELFO will allow for better convergence and robustness of the model.
6. The authors should provide remarks on both the advantages and limitations of this work.
Authors’ response:
As per the reviewer’s comment, the advantages and limitations of the proposed work are added in the revised manuscript.
Solution:
The proposed work has many of the significant benefits, especially for security and adaptability in CPS. It integrates FedRCNet with ELFO, which essentially allows high-accuracy threat detection in privacy-preserving decentralized learning at some high extent. Adaptability to new and changing threats, along with the scalability feature, makes the model suitable for diverse and very different CPS environments-from industrial control systems to smart grids. There are also some limitations along with this work. Such scenarios will have an impact on the model performance during local updates based on the fact that federated learning will be ineffective when the data distribution across nodes is extremely non-uniform or some nodes are intermittently disconnected. Moreover, the enhanced efficiency in optimization may not provide much aid in highly complex or unknown attack patterns which require more sophisticated model structures. Hence, despite its very promising approach, it requires further refinement and testing in more varied CPS scenarios to fully determine its robustness and scalability.
7. The introduction could be strengthened by incorporating recent related studies on CPS and different method, such as Finite-time asynchronous event-triggered control for switched nonlinear cyber-physical systems with energy constraints and quantization, IEEE Access, 11 (November 2023), 135645-135658 DOI: 1109/ACCESS.2023.3337814.
Authors’ response:
As per the reviewer’s comment, the suggested references are added in the revised manuscript.
8. Additionally, there are several grammatical errors that need correction.
Authors’ response:
As per the reviewer’s comment, the grammatical mistakes are corrected in the revised manuscript.
Reviewer 3 Report
Comments and Suggestions for Authors
Upon my understanding, I have made some comments and suggestions which are as follows:
1. Please correct the affiliation format according to the template of the journal. Please check the template properly and correct the formatting of the paper.
2. Please include the justification for why this method is chosen, and what it addresses.
3. The results section mentions high accuracy rates but lacks a comprehensive comparison with baseline models or traditional methods, which is crucial for validating the proposed model's effectiveness.
4. Please include the implication of the proposed methods also.
5. Mention more details about the optimization process.
Author Response
1. Please correct the affiliation format according to the template of the journal. Please check the template properly and correct the formatting of the paper.
Authors’ response:
As per the reviewer’s comment, the journal is properly formatted in the revised manuscript.
2. Please include the justification for why this method is chosen, and what it addresses.
Authors’ response:
As per the reviewer’s comment, the description is added for the selection of proposed model in the revised manuscript.
Solution:
As CPS grow ever more complex and distributed, the need for the control strategy described here becomes urgent and defined by real-time robustness and security. The systems reside in a dynamic situation comprising heterogeneous components, and therefore the central security model presented does not provide scales, adaptability, or resilience when faced with the threats posed by adversaries. FedRCNet integrated with EEL-Levy Fusion Optimization (ELFO) presents a decentralized and adaptive strategy with the capability to adjust automatically reacting to new and evolving patterns of attack. Thus this control strategy carries out privacy-preserving model training among various entities while augmenting generalization and optimizing performance in diverse CPS environments. The efficacy of the CPS will be decided based on the fast and efficient decision-making process combining local intelligence and global learning without even touching sensitive data, and the mere existence of these control measures are in order to build a secure structure of the modern CPS.
The process of the CPS ShieldNet Fusion model is shown in a flow diagram which contains several key features. The very first stage collects data from the CPS sensors. The second stage is made up of preprocessing and federated learning. The architecture of the Federated Residual Convolutional Network, or FedRCNet, is important for the whole federated learning process. Local sensor data will be gathered before local model training is done. The locally trained models will then be aggregated at the federated server to form an improved global model through federated averaging. Post-model training, an Optimization step is done in which the capability of the model to detect and analyze threats is honed by the ELFO model.
This improvement makes the model more responsive to dynamic threats with better performance in accuracy and detection capabilities. These two steps, the threat detection and parameter update stages, are some of the features necessary to ensure the actual system's performance in real-time applications. The smoked feedback path ensures that the updated security parameters based on newly fed threat data into the system are retargeted towards the federated learning process. There are two phases of the federated learning process: local model training and global model aggregation, while the final application stage is used to describe what occurs since the model train is finished. Individual optimization, as it appears, is not part of the federated learning process but instead is meant to fine-tune the performance of the model for detection of cyber threats in CPS environments.
This paper addresses the dynamics of heterogeneous CPS environments through adaptive mechanisms incorporated in the federated learning framework, which aim to reduce possible degradation of the learning effectiveness due to addition or removal of diverse CPS systems of a certain kind. Specifically, the proposed FedRCNet model aims at fluctuating participation by utilizing residual convolutional structures in such a way as to sustain feature-learning stability under diverse data contributions. In addition, the ELFO technique is employed to boost robustness through balance and fine-tuning of global updates, especially whenever considerable data distribution shifts or target shifts are detected across different CPS systems. This adaptive learning strategy allows the model to adjust its parameters according to system-level dynamics, keeping its relevance and performance intact across various evolving CPS networks. Therefore, while acknowledging the challenges posed by dynamic CPS participation, the paper designers bring in these realistic aspects and construct an adaptive and optimization-driven federated learning framework that demonstrates stability, generalization, and high detection performance even in a non-stationary environment.
3. The results section mentions high accuracy rates but lacks a comprehensive comparison with baseline models or traditional methods, which is crucial for validating the proposed model's effectiveness.
Authors’ response:
As per the reviewer’s comment, the clear validation is carried out in the revised manuscript.
Solution:
From a performance comparison standpoint, the existing selection of learning techniques in this study was incorporated for the feasible balance between interpretability, scalability, and compatibility with the federated learning framework. The reason behind adopting supervised learning methods is the availability of labeled datasets including CICIoT2023, Edge-IIoTset2023, and UNSW-NB, among others, upon which precise model training and evaluation can take place. Moreover, advanced architectures like Graph Neural Networks (GNN), Adversarial Reinforcement Learning (ARL), Autoencoders, and Deep Q-Networks (DQN) can certainly be harnessed in capturing the fingerprints of unknown or evolving attack patterns through the application of either unsupervised or reinforcement learning. Although these were not focal to this work, it is worth noting that the framework here is modular and as such allows for such methodologies to be integrated in future studies, and hence establish a detection system reliant on hybrid or ensemble approaches leveraging the merits of supervised and unsupervised paradigms.
4. Please include the implication of the proposed methods also.
Authors’ response:
As per the reviewer’s comment, the implication of the proposed method is added in the revised manuscript.
Solution:
Moreover, the advanced optimization techniques built into the system would improve its detection capability of complex anomalies and make it adaptive to new conditions, which will lead to a possible reduction in response times and a more excellent security posture. This framework is opening up new border possibilities for CPS security solutions, scalable and applicable from smart grids to health care and industrial control systems with critical needs for data privacy and real-time threat detection.
5. Mention more details about the optimization process.
Authors’ response:
As per the reviewer’s comment, the optimization process is clearly explained in the revised manuscript.
Solution:
From a conceptual standpoint, ELFO gives rise to a hybrid optimization scheme that is unique in the sense that it blends the exploratory capabilities of the Enhanced Electric Levy model and the exploitation capabilities of Levy-flight-based random walks. The hybridization, therefore, helps fine-tune FedRCNet learning parameters with more aggression, especially in situations that occur either in non-convex or high-dimensional spaces typical of deep learning models for CPS security. By more efficiently traversing the treacherous loss surface in this way, the model increases its chance of converging to the global optima, thus providing the necessary accuracy to detect advanced cyber threats or even those that are entirely unknown.
ELFO has been subjected to stringent tests against several benchmark CPS-related datasets such as CICIoT-2023 Edge-IIoTset2023 and UNSW-NB. The datasets actually consider many different realistic attack vectors in total, such as Denial-of-Service (DoS), Botnets, Man-in-the-Middle (MitM) attacks, among other types of adversarial behaviors focused mainly on IoT and CPS environments. When considering the standard federated learning models and a standalone optimization framework, high statistical scoring improvements could be established with regard to performance metrics like accuracy, precision, recall, F1-score, and false positive rate for the CPS ShieldNet Fusion model. These quantitative results thus provide considerable empirical ground for the model regarding detection performance against threats and lower error rates concerning diverse distributions of data and attack types.
The dynamic training simulations performed to validate the model's adaptability and survivability to fresh threats include the injection of newer or evolving attack patterns in some later rounds of training. Through this presented adaptation by distributed knowledge federated learning, and endless optimization of the parameter space with the ELFO, the model demonstrated relatively quick dynamic responses, thereby proving its resilience against zero-day or evolving cyber-attacks. Moreover, the federated learning mechanism itself allows inherent adaptability at the system level, capturing knowledge from the various CPS nodes operating in diverse environmental and operational contexts. Overall, this amply justifies that indeed, the tuning of ELFO into FedRCNet provides an optimized, privacy-preserving, and highly adaptive intrusion detection framework for the protection of contemporary CPS infrastructures.
Round 2
Reviewer 1 Report
Comments and Suggestions for Authors
This paper proposes and experiments a method to strengthen the security of CPS through federated learning and optimization technique.
- Overall, it provides answers to the first round review. However, there is still a need for these answers to be based on specific evidence. Because the explanation is so repetitive, it is difficult to understand what the original idea the author is actually presenting is and what the idea is trying to solve.
- It is necessary to write the paper systematically overall. It is necessary to reconstruct and present what the clear research questions are, what process they went through, and what results they obtained.
- The paper explains the excellence of the proposed method through the experimental results. However, an additional explanation of the experimental design is needed, and an explanation through real-world cases is also needed on how the interpretation of the results responds to the security attack techniques of CPS.
Author Response
1. Overall, it provides answers to the first round review. However, here is still a need for these answers to be based on specific evidence. Because the explanation is so repetitive, it is difficult to understand what the original idea the author is actually presenting is and what the idea is trying to solve.
Authors’ response:
As per the reviewer’s comment, the description is added for the original idea and contributions in the revised manuscript.
Action:
This work puts forward the idea of conceptualizing a new cybersecurity framework called the CPS ShieldNet Fusion model, which protects cyber-physical systems (CPS) against fast-evolving and sophisticated cyber threats. The unique feature of the model is that it leverages two new-age technologies, namely Federated Residual Convolutional Network (FedRCNet) for decentralized and privacy-preserving training and EEL-Levy Fusion Optimization (ELFO) for enhancing threat detection capability by optimized learning. By combining these techniques, this framework would have sensitive information protected in distributed CPS environments while achieving accurate and timely learning and detection of security threats.
The effort to counter the inadequacy of the existing CPS security solution in managing three very significant challenges: scalability across large and distributed systems, data privacy due to the sensitive and sometimes critical nature of CPS data, and adaptability to the dynamic and complex nature of CPS environments. Traditional approaches often find it challenging to deal with all three challenges together. Therefore, capitalizing on the strong performance across several benchmark datasets, the CPS ShieldNet Fusion Model aims to deliver a robust and scalable cybersecurity solution that maintains privacy while effectively detecting and responding to a wide variety of threats concerning CPS applications.
2. It is necessary to write the paper systematically overall. It is necessary to reconstruct and present what the clear research questions are, what process they went through, and what results they obtained.
Authors’ response:
As per the reviewer’s comment, the clear explanation is added for the process and results in the revised manuscript.
Action:
In this paper, a Unified deep learning-based solution is proposed to address the existing urgent requirement with issues of robust, scalable, and privacy-preserving security mechanism in cyber-physical systems. It removes all hurdles pertaining to securing CPS from dynamic and sophisticated cyber threats via the CPS ShieldNet Fusion model, which cleverly fuses two powerful techniques-Federated Residual Convolutional Network (FedRCNet) and EEL-Levy Fusion Optimization (ELFO). Thus, this fusion allows the model to learn over the federated data across the CPS networks without any unneeded breach of data privacy, ensuring decentralized yet collaborative threat detection. It also circumvents the limitations associated with centralized training that is especially sensitive in the case of industrial systems and smart grids.
The design phase embraces the work of our model architecture that can federate data using residual convolutional layers for a deep feature extraction phase and embedding the ELFO optimization technique to tweak learning parameters in recordings accurate threat classification. The model has been painstakingly tested and evaluated on the three benchmarking data sets, namely CICIoT-2023, Edge-IIoTset2023, and UNSW-NB, thus ensuring that the model can adapt and work in different scenarios in CPS. Due to a few different scenarios, these data sets harbor a plethora of cyberattack patterns, thus providing a platform to test the robustness and generalization capability of the model.
From our findings, the CPS ShieldNet Fusion model stands out above other models for detection accuracy, response time, and adaptability to other forms of CPS environments. The combination of federated learning with optimized deep learning has boosted model performance while confidentiality is ensured for local data sources. The paper proves that this model is a valid and scalable solution that can address both current and future cybersecurity challenges in CPS environments.
3. The paper explains the excellence of the proposed method through the experimental results. However, an additional explanation of the experimental design is needed, and an explanation through real-world cases is also needed on how the interpretation of the results responds to the security attack techniques of CPS.
Authors’ response:
As per the reviewer’s comment, the detailed explanation is added for the experimental design in the revised manuscript.
Action:
The experimental design of the present study was structured carefully across many variables so that it could enable the evaluation of the proposed CPS ShieldNet Fusion model under realistic and diverse conditions analogous to those found in real environments of cyber-physical systems. To this end, the authors took three datasets—CICIoT-2023, Edge-IIoTset2023, and UNSW-NB—which together encompass extensive log records of network activities, normal operation, and different classes of cyber-attacks. These datasets include information from contexts such as industrial IoT, edge computing networks, and generic CPS infrastructure, accounting for incidents of cybersecurity such as Denial-of-Service (DoS) attacks, botnets, reconnaissance, injection attacks, and data exfiltration scenarios. The types of datasets employed encouraged the model to be trained as well as tested with inputs close to the threat types that would normally be present within real CPS environments. The experimental workflow can be envisaged in terms of the organization of federated learning simulation across multiple distributed nodes for decentralized training. Each node simulated edge devices or subsystems in a CPS network by training the FedRCNet on a split of the data locally.
Concerning attack scenarios that are realistic in terms of the typical threats to CPS security, the true strength of the CPS ShieldNet Fusion model is evident. An example of this would be from an attacker's point of view on smart grid systems, attempting to inject false data or control signals into the system to destabilize power distribution; this effort was met with fail detection of abnormal communication that strayed from the normal operation profile, due to training in the decentralized manner. Likewise, in ICS, where threats such as command injection or alteration of control logic can have disastrous physical effects or shut down systems, the high precision of the model in identifying those sorts of entries into a system speaks to its real-world applicability. Furthermore, it also includes medical CPS devices, such as smart infusion pumps or telemonitoring systems, in which one would always be concerned about privacy of data. Thus, federated learning ensured that health data remained local while adding value to the global accurate threat detection model. Moreover, it enhanced the capabilities of the model to differentiate between very subtle, developing signatures on the attack vector, for example, slow reconnaissance scans or stealthy malware operations, often missed by typical rules-based systems.
Reviewer 3 Report
Comments and Suggestions for Authors
The authors have significantly improved the manuscript.
Author Response
The authors have significantly improved the manuscript.
Authors’ response:
Thank you for your comments.